# Watermarking for Out-of-distribution Detection

**Qizhou Wang**[1][*]  **Feng Liu**[2][*]  **Yonggang Zhang**[1]  **Jing Zhang**[3]
**Chen Gong**[4,5]  **Tongliang Liu**[6][†]  **Bo Han**[1][†]

[1]Department of Computer Science, Hong Kong Baptist University
[2]School of Mathematics and Statistics, The University of Melbourne
[3]School of Computer Science, The University of Sydney
[4]PCA Lab, Key Lab of Intelligent Perception and Systems for High-Dimensional Information of MoE
[5]Jiangsu Key Lab of Image and Video Understanding for Social Security,
School of Computer Science and Engineering, Nanjing University of Science and Technology
[6]TML Lab, The University of Sydney
{csqzwang, csygzhang, bhanml}@comp.hkbu.edu.hk
fengliu.ml@gmail.com    chen.gong@njust.edu.cn
{jing.zhang1, tongliang.liu}@sydney.edu.au

## Abstract

*Out-of-distribution* (OOD) detection aims to identify OOD data based on representations extracted from well-trained deep models. However, existing methods largely ignore the *reprogramming* property of deep models and thus may not fully unleash their intrinsic strength: *without* modifying parameters of a well-trained deep model, we can reprogram this model for a new purpose via data-level manipulation (e.g., adding a specific feature perturbation to the data). This property motivates us to reprogram a classification model to excel at OOD detection (a new task), and thus we propose a general methodology named *watermarking* in this paper. Specifically, we learn a unified pattern that is superimposed onto features of original data, and the model's detection capability is largely boosted after watermarking. Extensive experiments verify the effectiveness of watermarking, demonstrating the significance of the reprogramming property of deep models in OOD detection. The code is publicly available at: github.com/qizhouwang/watermarking.

## 1 Introduction

Deep learning systems in an open world often encounter *out-of-distribution* (OOD) inputs whose label spaces are disjoint with that of training data, known as *in-distribution* (ID) data. For safety-critical applications, deep models should make reliable predictions for ID data, meanwhile detecting OOD data and avoiding making predictions for the detected ones. This leads to the OOD detection task [28, 38, 41, 50], which has attracted intensive attention in the real world.

Identifying OOD data remains non-trivial since deep models can be overconfident with them [40]. As a promising technique, the *classification-based* OOD detection [58] relies on various scoring functions derived by classification models well trained with ID data (i.e., well-trained models), taking those inputs with small scores as OOD cases. In general, the scoring functions can be defined by logit outputs [17, 33], gradients [21], and embedding features [28, 44]. Without interfering with the well-trained models or requiring extra computation, they exploit the inherent capability of models learned from only ID data. In general, these advantages can be critical in reality, where the cost of re-training is prohibitively high and the acquisition of true OOD data is very difficult [58].

---

[*]Equal contributions.
[†]Correspondence to Bo Han (bhanml@comp.hkbu.edu.hk) and Tongliang Liu (tongliang.liu@sydney.edu.au).

36th Conference on Neural Information Processing Systems (NeurIPS 2022).

Although promising progress has been achieved, previous works largely ignore the *reprogramming* property [9] of deep models: a well-trained model can be repurposed for a new task by a proper transformation of original inputs (e.g., a universal feature perturbation), without modifying any model parameter. For example, a model pre-trained on ImageNet [6] dataset can be reprogrammed for classifying biomedical images [49]. This property indicates the possibility of making a well-trained model adapt for effective OOD detection, motivating us to make the *first* attempt to investigate if the reprogramming property of deep models can help to address OOD detection, *i.e., can we reprogram well-trained deep models for OOD detection (a new task)?*

In this paper, we propose a novel method, **watermarking**, to *reprogram* a well-trained model by adding a watermark to original inputs, making the model can help detect OOD data well. The *watermark* has the same shape with original inputs, which is a static pattern that can be added for test-time inputs (cf., Figure 1). The pre-defined scoring strategy (e.g., the free energy scoring [33]) is expected to be enhanced, with an enlarged gap of OOD scores between the watermarked ID and OOD data (cf., Figure 2).

It is non-trivial to find the proper watermark due to our lack of knowledge about unseen OOD data in advance. To address the issue, we propose a learning framework for effective watermarking. The insight is to make a well-trained model produce high scores for watermarked ID inputs meanwhile regularize the watermark such that the model will return low confidence without perceiving ID pattern. In this case, the model will have a relatively high score for a watermarked ID input, while the score remains low for OOD data (cf., Figure 2). The reason is that the model encounters a watermarked input but not seeing any ID pattern. In our realization, we

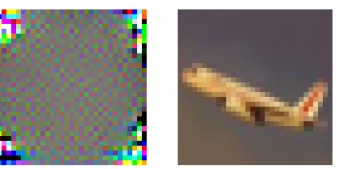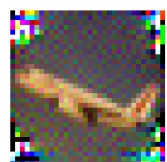

Figure 1: Watermarking on CIFAR-10 [26] with free energy scoring [33]. The left figure is the learned watermark; the middle figure is an original input; the right figure is the watermarked result.

adopt several representative scoring strategies, devising specified learning objectives and proposing a reliable optimization algorithm to learn an effective watermark.

To understand our watermarking, Figure 1 depicts the watermark learned on CIFAR-10 [26] dataset, with the free energy scoring [33]. As we can see, the centre area of the learned watermark largely preserves the original input pattern, containing the semantic message that guides the detection primitively. By contrast, the edge area of the original input is superimposed by the specific pattern of the watermark, which may *encode* the knowledge once hidden by the model in boosting OOD detection. Overall, watermarking can preserve the meaningful pattern of original inputs in detection, with the improved detection capability that is learned from the trained model and ID data.

Figure 2 demonstrates the effect of our learned watermark, which is an example with the free energy scoring. After watermarking, the scoring distributions are much concentrated, and the gap between ID (i.e., CIFAR-10) and OOD (i.e., SVHN [39] and Texture [4] datasets) data is enlarged notably. We conduct extensive experiments for a wide range of OOD evaluation benchmarks , and the results verify the effectiveness of our proposal.

The success of watermarking takes roots in the following aspects: (1) a well-trained model on classification has the potential to be reprogrammed for OOD detection since they are two related tasks; (2) reprogramming has been widely studied, ranging from image classification to time series analysis [6, 49], making our proposal general across various domains; and (3) OOD detection suffers from the lack of knowledge about the real-world OOD distributions. Fortunately, with only data-level manipulation in low dimensions, watermarking can largely mitigate this issue of limited data. Overall, this data-level manipulation is orthogonal to existing methods, and thus provides a new road in OOD detection and can inspire more ways to design OOD detection methods in the future.

## 2 Related Works

To begin with, we briefly review the related works in OOD detection and model reprogramming. Please refer to Appendix A for the detailed discussion.

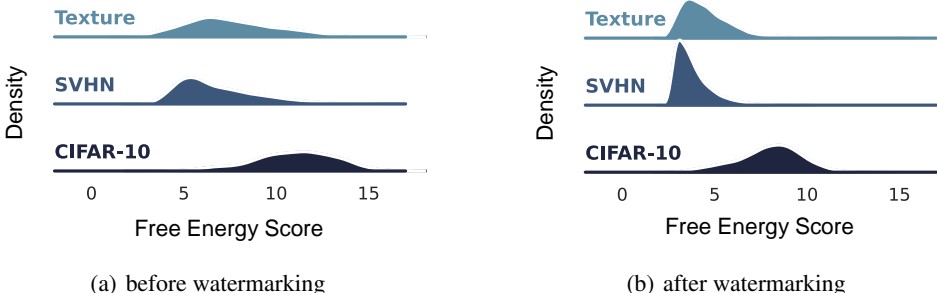

|  (a) before watermarking | (b) after watermarking |

Figure 2: Experimental results before (a) /after (b) watermarking with CIFAR-10 being the ID dataset, SVHN and Texture being the OOD datasets. Data with large (small) OOD scores should be taken as ID (OOD) data, and a larger distribution gap of scoring between ID and OOD data ensures a better detection performance. After watermarking, the gap between ID and OOD data is enlarged, demonstrating the improved capability of the original model in OOD detection. The horizontal axes are ignored for illustration, please refer to Figure 4 for a completed version.

**OOD Detection** discerns ID and OOD data by their gaps regarding the specified metrics/scores, and existing methods can be roughly divided into three categories [58], the *classification-based* methods, the *density-based* methods, and the *distance-based* methods. Specifically, the classification-based methods [17, 21, 33, 44] use representations extracted from the well-trained models in OOD scoring; and the distance-based methods [2, 20, 60] measure the distance of inputs from class centers in the embedding space. Moreover, the density-based methods estimate input density with probabilistic models [28, 41, 45], identifying those OOD data with small likelihood values. Distance-based and density-based methods may suffer from complexity in computation [28] and difficulty in optimization [62]. Therefore, more researchers focus on developing classification-based methods and have made big progress on benchmark datasets recently [21, 33].

**Model Reprogramming** repurposes well-trained models for new tasks with only data-level manipulation [9], indicating that deep models are competent for different jobs without changing any model parameter. In previous works, the data-level manipulation typically refers to a static padding pattern (different from our proposal) learned for the target task, which is added to the test-time data. The effectiveness of the model reprogramming is verified across image classification [9, 49] and time-series analysis [13, 57]. In this paper, we use the reprogramming property of deep models for effective OOD detection, which has been overseen previously.

## 3 Preliminary

Let $\mathcal{X} \subset \mathbb{R}^d$ be the input space and $\mathcal{Y} = \{1, \ldots, c\}$ be the label space. We consider the ID distribution $D_{\mathcal{X},\mathcal{Y}}^{\text{ID}}$ defined over $\mathcal{X} \times \mathcal{Y}$, the training sample $S_n = \{(\boldsymbol{x}_i, y_i)\}_{i=1}^n$ of size $n$ independently drawn from $D_{\mathcal{X},\mathcal{Y}}^{\text{ID}}$, and a classification model $f : \mathcal{X} \to \mathbb{R}^c$ (with logit outputs) well-trained on $S_n$.

Based on the model $f(\cdot)$, the goal of the classification-based OOD detection is to design a detection model $g : \mathcal{X} \to \{0, 1\}$ that can distinguish test-time inputs with the ID distribution $D_{\mathcal{X}}^{\text{ID}}$ from those with the OOD distribution $D_{\mathcal{X}}^{\text{OOD}}$. In general, $D_{\mathcal{X}}^{\text{OOD}}$ is defined as an irrelevant distribution of which the label set has no intersection with $\mathcal{Y}$, and thus should not be predicted by $f(\cdot)$. Overall, with $0$ denoting the OOD case and $1$ the ID case, the detection model $g(\cdot)$ is defined as

$$g(\boldsymbol{x}; \tau) = \begin{cases} 1 & s(\boldsymbol{x}; f) \geq \tau \\ 0 & s(\boldsymbol{x}; f) < \tau \end{cases}, \tag{1}$$

where $\tau \in \mathbb{R}$ is a threshold and $s : \mathcal{X} \to \mathbb{R}$ is the scoring function defined by $f(\cdot)$ whose parameters are fixed. Here, we focus on two representative methods in the classification-based OOD detection, namely, the *softmax scoring* and the *free energy scoring*.

**Softmax Scoring Function** [17] uses the maximum softmax prediction in OOD detection, of which the scoring function $s_{\text{SM}}(\cdot)$ is given by

$$s_{\text{SM}}(\boldsymbol{x}; f) = \max_k \texttt{softmax}_k \ f(\boldsymbol{x}), \tag{2}$$

where $\texttt{softmax}_k(\cdot)$ denotes the $k$-th element of the softmax outputs. In general, with a large (small) $s_{\text{SM}}(\boldsymbol{x}; f)$, the detection model will take the input $\boldsymbol{x}$ as an ID (OOD) case.

**Free Energy Scoring Function** [33] adopts the free energy function for scoring, defined by the logit outputs with the $\texttt{logsumexp}$ operation, namely,

$$s_{\text{FE}}(\boldsymbol{x}; f) = \log \sum_k \exp f_k(\boldsymbol{x})/T, \tag{3}$$

where $T > 0$ is the temperature parameter, fixed to 1 [33]. It aligns with the density of inputs to some extent, and thus is less susceptible to the overconfidence issue than the softmax scoring [33].

## 4 Watermarking Strategy

This section introduces the key concepts of watermarking for classification-based OOD detection.

**Definition.** A watermark $\boldsymbol{w} \in \mathbb{R}^d$ is a unified pattern with the exact shape as original inputs. It is added to test-time inputs statically, and we refer $\boldsymbol{w} + \boldsymbol{x}$ a *watermarked input* for $\forall \boldsymbol{x} \in \mathcal{X}$. In expectation, regarding the specified scoring function $s(\cdot)$, our watermarking should make the model excel at OOD detection for watermarked data.

**Learning Strategy.** Given the scoring function $s(\cdot)$, it is challenging to devise the exact watermark pattern by predefined rules. Therefore, for the proper watermarks in OOD detection, we need to devise learning objectives with respect to watermarks, which consider both ID and OOD data.

We generally have no information about the OOD distribution $D_{\mathcal{X},\mathcal{Y}}^{\text{OOD}}$, while we still want the model excels in discerning ID and OOD data from scoring. For this challenge, we make the model produce high scores if watermarked ID data are observed; meanwhile, we regularize the watermark such that the model will return low scores when ID patterns do not exist. From the lens of our model, the scores should remain low if a watermarked OOD input is given since the watermark is not trained to perceive OOD data, of which the patterns are very different from the ID data.

**Benefits of Watermarking.** Watermarking directly reprograms the model to make an adaptation to our specified task of scoring, such that the detection capability of the original model is largely improved. By contrast, previous methods typically adapt to their specified tasks by only the threshold $\tau$ as in Eq. (1). However, it requires the trade-off between false positive (ID) and false negative (OOD) rates when densities of scoring are non-separable (cf., Figure 2(a)).

Further, watermarking enjoys the benefits of previous classification-based methods in that we do not modify the original training procedure in classification, making our proposal easy to be deployed in real-world systems. Although the watermark also should be learned, the parameter space is in low dimension, and the learning procedure could be conducted *post-hoc* after the systems are deployed.

**Comparison with Existing Works**. In OOD detection, this paper is a first attempt in using the reprogramming property of deep models, leading to an effective learning framework named watermarking. At first glance, our methodology is seemingly similar to ODIN [32], which also conducts data-level perturbation for OOD detection. However, their instance-specified perturbation relies on extra backward-forward iterations during the test, which is not required in our method. Further, ODIN is designed for the softmax scoring, but our proposal is much general in OOD detection.

## 5 Realizations of Watermarking Strategy

In this section, we discuss our learning framework of watermarking in detail.

**Learning Objectives.** As mentioned above, we need to consider the ID and OOD situations separately, with the associated loss functions denoted by $\ell^{\text{ID}}(\cdot)$ and $\ell^{\text{OOD}}(\cdot)$. For the ID case, the ID training data are required, where we make the high scores for their watermarked counterparts. By contrast, since

we typically lack knowledge about the test-time OOD data, only the watermark is used here, and we expect the model to produce the score as low as possible when only perceiving the watermark.

Further, since only the watermark is adopted for training in the OOD case, the learned watermark is pretty sensitive regarding the detection model, i.e., the model may return different predictions when facing small perturbations. Thus, the watermarked OOD inputs may not guarantee the low scores. To this end, the watermark is further perturbed during training. Here we adopt the Gaussian noise, leading to the perturbed watermark of the form $\epsilon + w$ with $\epsilon \sim \mathcal{N}(\mathbf{0}, \sigma_1 \mathbf{I}_d)$ the *independent and identically distributed* (i.i.d.) Gaussian noise of $d$-dimension (the mean $\mathbf{0}$ and the standard deviation $\sigma_1 \mathbf{I}_d$). Then, the overall risk can be written as,

$$\mathcal{L}_n(\boldsymbol{w}) = \underbrace{\sum_n \ell^{\text{ID}}(\boldsymbol{x}_i + \boldsymbol{w}, y_i; f)}_{\mathcal{L}_n^{\text{ID}}(\boldsymbol{w})} + \beta \underbrace{\sum_n \ell^{\text{OOD}}(\boldsymbol{\epsilon}_j + \boldsymbol{w}; f)}_{\mathcal{L}_n^{\text{OOD}}(\boldsymbol{w})}, \tag{4}$$

with $\beta \geq 0$ the trade-off parameter, $\mathcal{L}_n^{\text{ID}}(\boldsymbol{w})$ the risk for ID data, and $\mathcal{L}_n^{\text{OOD}}(\boldsymbol{w})$ the risk for OOD data.

**Optimization.** To find the proper watermark, we use the first-order gradient update to iteratively update watermark's elements. However, data-level optimization remains difficult in deep learning, of which the results may get stuck at suboptimal points [52]. A common approach is to use the signum of first-order gradients, guiding the updating rule of the current watermark via

$$\boldsymbol{w} \leftarrow \boldsymbol{w} - \alpha \texttt{sign}(\nabla_{\boldsymbol{w}} \mathcal{L}_n(\boldsymbol{w})), \tag{5}$$

where $\texttt{sign}(\cdot)$ denotes the signum function and $\alpha > 0$ is the step size [35].

Further, for generality and insensibility, we prefer the solution that lies in the neighbourhood having uniformly low loss, i.e., with a smooth loss landscape [24]. Therefore, we adopt the *sharpness-aware minimization* (SAM) [10], an effective optimization framework in the seek of both the low loss value and the smooth loss landscape. Specifically, given the original risk $\mathcal{L}_n(\boldsymbol{w})$, the SAM problem is:

$$\mathcal{L}_n^{\text{SAM}}(w) = \max_{||\boldsymbol{\kappa}||_2 \leq \rho} \underbrace{[\mathcal{L}_n(\boldsymbol{w} + \boldsymbol{\kappa}) - \mathcal{L}_n(\boldsymbol{w})]}_{\text{sharpness}} + \mathcal{L}_n(\boldsymbol{w}) = \max_{||\boldsymbol{\kappa}||_2 \leq \rho} \mathcal{L}_n(\boldsymbol{w} + \boldsymbol{\kappa}) \tag{6}$$

where $\rho \geq 0$ is a constraint. For efficiency, the SAM makes the first-order Taylor expansion w.r.t. $\boldsymbol{\kappa}$ around $\mathbf{0}$, obtaining the approximated solution of the form [3]:

$$\boldsymbol{\kappa} = \rho \texttt{sign}(\nabla_{\boldsymbol{w}} \mathcal{L}_n(\boldsymbol{w})) \frac{|\nabla_{\boldsymbol{w}} \mathcal{L}_n(\boldsymbol{w})|^{q-1}}{(||\nabla_{\boldsymbol{w}} \mathcal{L}_n(\boldsymbol{w})||_q^q)^{1/p}}, \tag{7}$$

where $1/p + 1/q = 1$ and we set $p = q = 2$ for simplicity. Therefore, the estimation form of the SAM is written as $\mathcal{L}_n(\boldsymbol{w} + \boldsymbol{\kappa})$, with corresponding updating rule of

$$\boldsymbol{w} \leftarrow \boldsymbol{w} - \alpha \texttt{sign}(\nabla_{\boldsymbol{w}} \mathcal{L}_n(\boldsymbol{w} + \boldsymbol{\kappa})), \tag{8}$$

yielding an efficient optimization algorithm that induces the effective watermark.

**The Overall Algorithm.** In summary, we describe the overall learning framework. To begin with, the watermark is initialized by the i.i.d. Gaussian noise with the $\mathbf{0}$ mean and a small standard deviation $\sigma_2 \mathbf{I}_d$, and the learning procedure consists of three stages for each updating step:

- Negative sampling: a set of noise data $\epsilon$ is sampled, assuming be of the size $m$ as that of the mini-batch regarding the ID sample;
- Risk calculating: the risk for ID and OOD data are computed, and the overall risk is given by their sum with a trade-off parameter $\beta$ as in Eq. (4);
- Watermark updating: the first-order gradient guides the pixel-level update of the watermark, using the signum of gradients and the SAM to make a reliable update as in Eq. (8).

The learned watermark is added to test-time inputs for OOD detection, and the detection model with the pre-defined scoring function is then deployed. Appendix B summarizes our learning framework of watermarking. Moreover, two specifications of watermarking are discussed in the following.

---

[3]With an abuse of notation, we denote the estimated solution in the SAM as $\boldsymbol{\kappa}$ for simplicity.

**Two Realizations.** Here, we focus on two representative methods in OOD detection, namely, the softmax scoring and the free energy scoring. For other representative methods in OOD detection, please refer to Appendix C for their descriptions and the experiments.

*Softmax Scoring-based Watermarking.* Following [18], we set $\ell_{\text{SM}}^{\text{ID}}(\cdot)$ to be the cross entropy loss and $\ell_{\text{SM}}^{\text{OOD}}(\cdot)$ to be the cross entropy regarding the uniform distribution, namely,

$$\ell_{\text{SM}}^{\text{ID}}(\boldsymbol{x}, y; f) = -\log \texttt{softmax}_y f(\boldsymbol{x}) \ \text{ and } \ \ell_{\text{SM}}^{\text{OOD}}(\boldsymbol{x}; f) = -\sum_k \frac{1}{c} \log \texttt{softmax}_k f(\boldsymbol{x}), \quad (9)$$

specifying the learning objectives in Eq. (4) for the softmax scoring-based watermarking.

*Free-Energy Scoring-based Watermarking.* [33] use a set of learning objectives for model re-training with free energy scoring. However, their `logsumexp` operation originating from the free energy function is difficult for optimization, posing notorious computing issues [36]. To this end, we drop the log operation and make the overall risk always positive by the following learning objectives:

$$\ell_{\text{FE}}^{\text{ID}}(\boldsymbol{x}; f) = \sum_k \exp -f_k(\boldsymbol{x})/T_1 \ \text{ and } \ \ell_{\text{FE}}^{\text{OOD}}(\boldsymbol{x}; f) = \sum_k \exp f_k(\boldsymbol{x})/T_2, \quad (10)$$

realizing the learning objectives in Eq. (4) for the free energy scoring.

## 6 Experiments

In this section, we conduct extensive experiments for watermarking in OOD detection. Specifically, we demonstrate the effectiveness of our method on a wide range of OOD evaluation benchmarks; we conduct experiments for the important hyper-parameters in our learning framework; and we provide further experiments for an improved interpretation of our proposal.

Baselines results are achieved by our re-run of the publicly available codes. The source code of our proposal is released at github.com/qizhouwang/watermarking. All the methods are realized by Pytorch 1.81 with CUDA 11.1, where we use several machines equiped with GeForce RTX 3090 GPUs and AMD Ryzen Threadripper 3960X Processors.

**ID and OOD Datasets**. We use CIFAR-10, CIFAR-100 [26], and ImageNet [42] datasets as three ID datasets, with data pre-processing including horizontal flip and normalization. Furthermore, for the OOD datasets, we adopt several commonly-used benchmarks, including Textures [4], SVHN [39], Places365 [63], LSUN [59], and iSUN [56]. Referring to Appendix C.7 for hyper-parameter settings.

**Evaluation Metrics**. The performance in OOD detection is measured via three representative metrics, which are all threshold-independent [5]: (1) the false positive rate of OOD sample when true positive rate of ID data is at $95\%$ (FPR95); (2) the *area under the receiver operating characteristic curve* (AUROC), interpreted as the probability that an ID input has a greater score than an OOD one; and (3) the *area under the precision-recall curve* (AUPR), which further adjusts for different base rates.

**Configuration**. Following previous works [33], we employ WideResNet [61] (WRN-40-2) as the backbone model. For the CIFAR benchmarks, the models are trained for 200 epochs via the stochastic gradient descent, with the batch size 64, the momentum 0.9, and the initial learning rate 0.1. The learning rate is divided by 10 after 100 and 150 epochs. For the ImageNet Benchmark, the model is trained for 120 epochs via the stochastic gradient descent, with the batch size 32, the momentum 0.9 and the initial learning rate 0.05. The learning rate is divided by 10 after 60 and 90 epochs.

**CIFAR Benchmarks**. We depict the learned watermarks in Figure 3. As we can see, the centre areas maintain the pattern of original inputs, which are helpful in OOD detection primitively. By contrast, the edge areas of the watermarks distort the original features, superimposed with the pattern that may further boost the capability of the original models in OOD detection.

Then, we demonstrate the improvement of watermarking on CIFAR-10 and CIFAR-100 datasets. For the results of the softmax scoring in Table 1, our watermarking reduces the average FPR95 by $2.99 \sim 12.84$, boosts the average AUROC by $3.09 \sim 3.69$ and the average AUPR by $1.10 \sim 1.18$. Moreover, for each of the considered test-time OOD dataset, there also exist stable improvements after watermarking, except for the comparable results regarding Places365 on CIFAR-100 dataset.

For the results of the free energy scoring in Table 2, the improvement via watermarking is also substantial, with $4.54 \sim 13.98$, $3.43 \sim 4.65$, and $1.13 \sim 1.37$ better performance regarding the

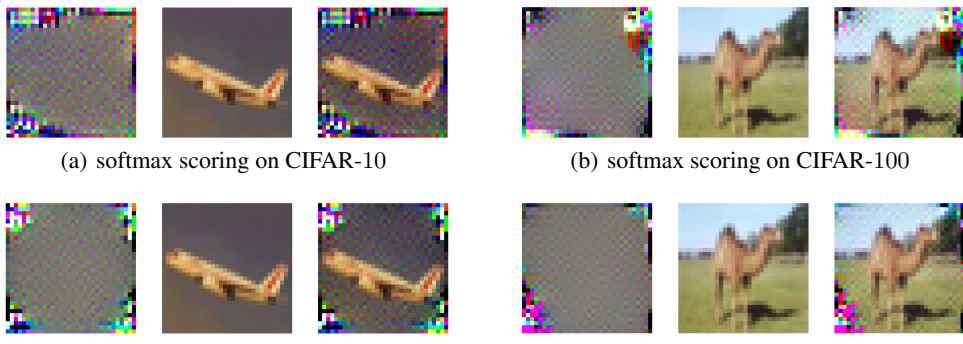



(a) softmax scoring on CIFAR-10      (b) softmax scoring on CIFAR-100

(c) free energy scoring on CIFAR-10      (d) free energy scoring on CIFAR-100



Figure 3: The learned watermarks (left) and the example images with (middle) and without (right) the watermarks. All the pictures are clamped between 0 and 255 for the purpose of illustration.

Table 1: Softmax scoring with/without watermarking on CIFAR benchmarks. ↓ (↑) indicates smaller (larger) values are preferred.

|  | FPR95 ↓ | AUROC ↑ | AUPR ↑ |
|---|---|---|---|
|  | w/ (w/o) watermark | | |
| **CIFAR-10** | | | |
| iSUN | **43.60** (55.55) | **93.53** (90.14) | **98.67** (97.84) |
| Places365 | **60.75** (62.50) | **87.85** (87.41) | **96.98** (96.94) |
| Texture | **42.00** (59.30) | **92.83** (88.37) | **98.43** (97.14) |
| SVHN | **27.25** (49.10) | **96.00** (91.69) | **99.17** (96.54) |
| LSUN | **40.70** (52.05) | **94.36** (91.50) | **98.86** (98.16) |
| **average** | **42.86** (55.70) | **92.91** (89.82) | **98.42** (97.32) |
| **CIFAR-100** | | | |
| iSUN | **77.85** (83.35) | **79.91** (75.28) | **95.35** (94.00) |
| Places365 | 83.25 (**82.20**) | 74.28 (**74.40**) | **93.47** (93.44) |
| Texture | **79.10** (83.80) | **77.14** (72.83) | **94.26** (92.81) |
| SVHN | **82.95** (85.05) | **76.92** (70.64) | **94.72** (92.61) |
| LSUN | **76.75** (80.45) | **79.60** (76.25) | **95.27** (94.32) |
| **average** | **79.98** (82.97) | **77.57** (73.88) | **94.61** (93.43) |

Table 2: Free energy scoring with/without watermarking on CIFAR benchmarks. ↓ (↑) indicates smaller (larger) values are preferred.

|  | FPR95 ↓ | AUROC ↑ | AUPR ↑ |
|---|---|---|---|
|  | w/ (w/o) watermark | | |
| **CIFAR-10** | | | |
| iSUN | **16.30** (32.10) | **96.97** (92.84) | **99.39** (98.33) |
| Places365 | **36.25** (41.45) | **91.87** (89.65) | **97.94** (97.21) |
| Texture | **32.60** (52.05) | **93.14** (85.43) | **98.08** (95.52) |
| SVHN | **16.45** (35.25) | **97.11** (90.91) | **99.39** (97.68) |
| LSUN | **16.85** (27.50) | **96.97** (93.98) | **99.38** (98.59) |
| **average** | **23.69** (37.67) | **95.21** (90.56) | **98.83** (97.46) |
| **CIFAR-100** | | | |
| iSUN | **75.05** (81.80) | **83.07** (79.04) | **96.15** (94.98) |
| Places365 | **80.45** (80.50) | **77.78** (74.99) | **94.45** (93.37) |
| Texture | **75.15** (80.20) | **79.55** (76.00) | **94.79** (93.53) |
| SVHN | **82.85** (85.10) | **75.26** (74.20) | **94.18** (93.70) |
| LSUN | **71.85** (80.45) | **84.01** (78.29) | **96.33** (94.69) |
| **average** | **77.07** (81.61) | **79.93** (76.50) | **95.18** (94.05) |

average FPR95, AUROC, and AUPR. Overall, Table 1 and Table 2 not only justify the effectiveness of watermarking and also demonstrate the generality of the proposed watermarking. Further, comparing between CIFAR-10 and CIFAR-100 datasets, the improvements of watermarking on CIFAR-10 is much greater than that of CIFAR-100, aligning with the previous observations [22] that a large semantic space can exaggerate the challenge in effective OOD detection.

Figure 4 illustrates the scoring distributions before (a) and after (b) watermarking on CIFAR-10, where we take the free energy scoring as an example. Due to the space limit, we only consider two test-time OOD datasets, namely, Texture and SVHN. As we can see, after watermarking, the distribution gap between the ID (i.e., CIFAR-10) and the OOD (i.e., Texture and SVHN) data is enlarged, and thus the detection capability is improved.

**ImageNet Benchmark.** Huang and Li [22] show that many advanced methods developed on the CIFAR benchmarks can hardly work for the ImageNet dataset due to its large semantic space with 1k classes. In order to verify the power of watermarking in large semantic space, we conduct experiments with ImageNet being the ID dataset, and the results regarding the softmax scoring and the free energy scoring are summarized in Table 3 and Table 4. For the softmax scoring, we decrease the average FPR95 from 54.93 to 40.50 after watermarking; for the free energy scoring, we reduce the average FPR95 from 52.73 to 43.23. It demonstrates that our watermarking still works well in the case of the large semantic space, with considerable improvements in OOD detection power.

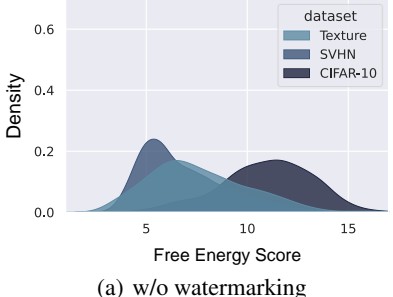

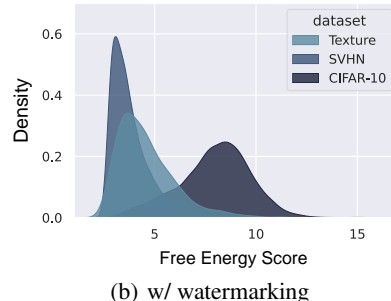

(a) w/o watermarking        (b) w/ watermarking

Figure 4: An illustration on CIFAR-10 dataset regarding the free energy scoring. (a) depicts the scoring distributions before watermarking, and (b) is the scoring distributions after watermarking.

Table 3: The softmax scoring with/without watermarking on ImageNet. The notion ↓ (↑) indicates smaller (larger) values are preferred.

|  | FPR95 ↓ | AUROC ↑ | AUPR ↑ |
|---|---|---|---|
|  | w/ (w/o) watermark | | |
| iSUN | **11.54** (52.45) | **97.41** (92.52) | **99.45** (98.70) |
| Places365 | **70.59** (73.25) | **82.03** (80.78) | **95.62** (94.58) |
| Texture | **61.20** (67.18) | **84.00** (82.27) | **98.60** (97.85) |
| SVHN | 44.58 (**28.49**) | 93.56 (**95.60**) | 98.70 (**99.00**) |
| LSUN | **11.84** (54.62) | **97.97** (91.52) | **99.57** (98.25) |
| average | **40.50** (54.93) | **91.22** (88.57) | **98.42** (97.69) |

Table 4: The free energy scoring with/without watermarking on ImageNet. The notion ↓ (↑) indicates smaller (larger) values are preferred.

|  | FPR95 ↓ | AUROC ↑ | AUPR ↑ |
|---|---|---|---|
|  | w/ (w/o) watermark | | |
| iSUN | **32.83** (45.40) | **94.35** (94.00) | **98.89** (98.20) |
| Places365 | **71.85** (75.01) | **79.85** (78.54) | **94.65** (94.40) |
| Texture | **67.75** (68.77) | **80.80** (80.22) | **97.00** (96.51) |
| SVHN | **12.85** (27.60) | **97.68** (95.17) | **99.45** (99.00) |
| LSUN | **33.75** (46.47) | **93.71** (90.59) | **98.80** (97.94) |
| average | **43.23** (52.73) | **89.10** (86.14) | **97.73** (97.15) |

**Near OOD Detection.** The above experiments focus on the far OOD detection setups where ID and OOD data are different regarding semantics and styles. Now, we further demonstrate the power of our watermarking strategy in a near OOD situation [54], covering a challenging situation where ID and OOD data have similar styles (i.e., near OOD data). Except for the common learning setup in Section 5, we further consider the use of shifting augmentations [47], which are data augmentations that are harmful to the standard contrastive learning but can be used to construct near OOD data similar to ID data. We consider two representative shifting augmentations: "permute" (permute evenly partitioned data) and "rotate" (rotate 90 degrees of original data). The shifting-augmented ID data are taken as OOD data fed into $\ell^{\text{OOD}}(\cdot)$ along with random Gaussian noise.

Table 5: The performance of our watermarking on near OOD detection regarding the softmax scoring and the free energy scoring.

|  | FPR95 | AUROC | AUPR |
|---|---|---|---|
| softmax scoring | | | |
| w/o watermark | 90.10 | 55.47 | 86.16 |
| common | 88.25 | 53.16 | 84.75 |
| permute | 86.45 | 60.04 | 86.33 |
| rotate | **81.50** | **65.69** | **88.67** |
| free energy scoring | | | |
| w/o watermark | 52.25 | 86.49 | 96.44 |
| common | 49.75 | 88.52 | 96.98 |
| permute | 48.55 | 88.40 | 97.03 |
| rotate | **47.85** | **88.90** | **97.08** |

The near OOD experiments are summarized in Table 5, where we take CIFAR-10 as ID data and CIFAR-100 as OOD data. Here, the common learning setup (common) already leads to improved performance compared to the cases without watermarking (w/o watermark). Moreover, watermarking with shifting augmentations (permute and rotate) can further boost the detection power of the models, leading to at most 8.60 and 4.70 improvements in FPR95 for the softmax and the free energy scoring.

**Effect of Hyper-parameters.** To further interpret our learning framework, we compare the performance of the learned watermarks regarding various setups of hyper-parameters, focusing on the standard deviation $\sigma_1$ in the Gaussian noise and the perturbation constraint $\rho$ in the SAM, which are both critical. As a case study, we conduct experiments regarding the softmax scoring on CIFAR-10. For detailed results about the ablation study, please refer to Appendix C.

Table 6 lists the results with various values of $\sigma_1$, ranging from 0.00 to 2.00. Note that $\sigma_1$ is the standard deviation of the Gaussian noise added to the watermark, with different values indicating

Table 6: The average performance of the softmax scoring on CIFAR-10 dataset with various values of the parameter $\sigma_1$. The notion ↓ (↑) indicates smaller (larger) values are preferred.

| $\sigma_1$ | FPR95 ↓ | AUROC ↑ | AUPR ↑ |
|---|---|---|---|
| 2.00 | 42.11 | 92.01 | 98.24 |
| 1.60 | 41.41 | 92.14 | 98.25 |
| 1.20 | 41.98 | 91.91 | 98.20 |
| 0.80 | 43.38 | 91.89 | 98.21 |
| 0.40 | **38.66** | **93.03** | **98.45** |
| 0.00 | 48.71 | 91.43 | 98.11 |

Table 7: The average performance of the softmax scoring on CIFAR-10 dataset with various values of the parameter $\rho$. The notion ↓ (↑) indicates smaller (larger) values are preferred.

| $\rho$ | FPR95 ↓ | AUROC ↑ | AUPR ↑ |
|---|---|---|---|
| 5.00 | 60.02 | 87.36 | 97.15 |
| 1.00 | **39.12** | **92.96** | **98.42** |
| 0.50 | 43.55 | 92.38 | 98.34 |
| 0.10 | 41.99 | 92.77 | 98.41 |
| 0.05 | 42.06 | 92.84 | 98.42 |
| 0.00 | 43.04 | 92.44 | 98.32 |

various degrees of the perturbation. As we can see, a mild perturbation of the watermark (e.g., $\sigma_1 = 0.40$) can truly lead to improved results in detection, with 10.05, 1.60, and 0.34 improvements regarding the average FPR95, AUROC, and AUPR. It indicates that the learned watermark is sensitive when facing small perturbations, if the Gaussian noise is not applied during training. However, some extreme values (e.g., $\sigma_1 = 2.00$) may overwhelm the watermark pattern and thus be detrimental.

In Table 7, we summarize the experimental results given by various values of $\rho$ for the SAM. Overall, a large value of $\rho$ indicates that a wide range in the solution's neighbours should be smooth, and thus the stability of the result is expected to be improved. However, such a solution may consume too much capacity of the watermark, misleading the learning procedure to some unsatisfactory results. On the other side, our watermarking truly benefits from the SAM with a mild choice of the hyper-parameter. Specifically, comparing with the results without the SAM (i.e., $\rho = 0.00$), the detection capability of the watermark with a suitable $\rho$ (i.e., $\rho = 1.0$) is largely improved, with 3.92, 0.52, and 0.10 better results regarding the average FPR95, AUROC, and AUPR.

**Transferability of watermarking**. Further, we explore the transferability of the watermarking strategy learned with different scoring functions, e.g., we study the effect of the watermark learned with the softmax scoring when it is deployed with the free energy scoring. The results are summarized in Table 8. Note that, a "learn" with FE and "score" with SM indicates that the watermark is learned with the free energy scoring and tested regarding the softmax scoring. As we can see, the watermarks learned with the free energy scoring can be reused for the softmax scoring (even with better results), while the reverse leads a deterioration. It indicates that our learning objectives in Eq. (10) are general and effective for both the softmax scoring and the free energy scoring. However, we do not observe any transferability between different datasets. For example, for the softmax scoring, when the watermark learned on CIFAR-10 is adopted for CIFAR-100, there is a drop of performance with 51.93 (from 42.86 to 94.79), 35.29 (from 92.91 to 57.62) and 12.00 (98.42 to 86.42) regarding the average FPR95, AUROC, and AUPR. Further studies may be required here.

Table 8: Transferability of watermarking across scoring functions. SM denotes softmax scoring and FE denotes free energy scoring.

| learn | score | FPR95 ↓ | AUROC ↑ | AUPR ↑ |
|---|---|---|---|---|
| | | CIFAR-10 | | |
| SM | SM | 42.86 | 92.91 | 98.42 |
| FE | SM | **40.19** | **94.83** | **98.99** |
| FE | FE | **23.69** | **95.21** | **98.83** |
| SM | FE | 28.22 | 94.70 | 98.81 |
| | | CIFAR-100 | | |
| SM | SM | 79.98 | **77.57** | **94.61** |
| FE | SM | **77.07** | 76.27 | 94.02 |
| FE | FE | **77.07** | **79.93** | **95.18** |
| SM | FE | 78.48 | 79.24 | 94.56 |

# 7 Conclusion

This paper demonstrates that the model's inherent capability in OOD detection can be largely improved with only data-level manipulation, where we propose a general and effective methodology named watermarking. Overall, we learn a discriminative pattern that could be superimposed onto original inputs, such that the OOD scores become much separable between ID and OOD data. Our results indicate a promising direction in OOD detection that warrants our exploration in the future. Further investigation should focus on the stability in optimization and general learning objectives regarding advanced OOD detection methods.

## Acknowledgments and Disclosure of Funding

QZW, YGZ and BH were supported by the RGC Early Career Scheme No. 22200720, NSFC Young Scientists Fund No. 62006202, Guangdong Basic and Applied Basic Research Foundation No. 2022A1515011652, RGC Research Matching Grant Scheme No. RMGS2022_11_02 and No. RMGS2022_13_06, and HKBU CSD Departmental Incentive Grant. CG was supported by NSF of China No. 61973162, NSF of Jiangsu Province No. BZ2021013, NSF for Distinguished Young Scholar of Jiangsu Province No. BK20220080, and the Fundamental Research Funds for the Central Universities No. 30920032202 and No. 30921013114. TL was partially supported by Australian Research Council Projects DP180103424, DE-190101473, IC-190100031, DP-220102121, and FT-220100318.

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
