# A Additional Related Works

We review the recent studies in OOD detection, model reprogramming, and backdoor attack.

## A.1 OOD Detection

Following [58], we attribute existing works into three categories, namely, the *classification-based* methods, the *density-based* methods, and the *distance-based* methods. In general, these methods aim to maximize the gap between ID and OOD data regarding specified metrics in identifying OOD data.

**The classification-based methods** use the representations extracted from the well-trained classification models in OOD scoring. For example, [17, 32, 33, 37, 46, 51] employ logit outputs from models in estimating the confidence of ID data; [28, 44] adopt Mahalanobis distance and Gram Matrix to exploit models' detection capability from embedding features; [21, 32] further demonstrate the importance of gradient information, either perturbing inputs with its gradients or directly using the gradient norm in scoring. The classification-based methods are easy to be deployed without modifying the models [58], and thus it is the main focus in this paper.

**The distance-based methods** measure the distance regarding the embedding space, taking those data far away from the class prototypes as the OOD data. Representative works adopt the Mahalanobis distance [21, 28], the cosine similarity [2, 60], and the Euclidean distance [20]. Our methods can also be used in the distance-based methods. However, extra computation, such as calculating the precision matrix [28], may lead difficulty in devising proper learning objectives, out of the scope of our paper.

**The density-based methods** explicitly estimate the density of ID samples with various probabilistic models, identifying those OOD data based on the likelihood [28], the likelihood ratio [31, 41, 45], and the likelihood regret [55]. Typically, the input density is modelled by the mixture of Gaussian models [28] and the flow-based methods [25, 38]. Although the density-based methods can directly characterize the properties of ID density, these methods are difficult to be trained and may make overconfident predictions, as demonstrated in previous works [37, 38].

Recent works also focus on the causes of challenges in OOD detection, from the lens of the BatchNorm statistics [46], the density estimation [37], and the spurious correlation [44]. Improved methods, related to specified model architectures [7, 53], data perturbation [1, 3, 32], data augmentation [15, 19, 48], and outlier exposure [18, 23], are also well-studied. However, these methods typically overlook the reprogramming property of deep models, which remain orthogonal to our proposal.

## A.2 Reprogramming Property

The seminal work [9] introduces adversarial reprogramming as an attack method in adversarial learning [11], adversarially reprogramming the target model to perform a new task without changing the original model. The term "attack" lies in the fact that, by reprogramming, an attacker can easily steal public machine learning services, abusing their computational resources for tasks that violate their original purposes. Overall, [9] claim the reprogramming property of deep models—without modifying parameters of a well-trained model, we can reprogram it for a new purpose with only data-level manipulation. The reprogramming property of deep learning is preliminarily verified for various tasks, and its further applications are not limited to adversarial learning. Actually, advanced works often take reprogramming as an effective transfer learning technique in the cases with limited data and computing resources. In the context of the image classification, [9] reprogram a model trained on ImageNet in solving vision-based counting tasks; and [49] further consider reprogramming a black-box system for biomedical image classification, which suffers from label scarcity issue. In the context of the natural language processing, well-trained models are reprogrammed for time-series classification [57] and sentiment analysis [13], where data scarcity issues frequently occur.

In this paper, we also employ the reprogramming property of deep models for transfer learning. However, instead of reprogramming across different datasets, we reprogram our original classification model for the task in OOD detection, considering the situation with the same (ID) dataset before/after reprogramming. Further, to preserve the benefits of previous classification-based detection methods, we adopt the perturbing pattern (i.e., the watermark) on the same shape as the original inputs, instead of reshaping original inputs and adding padding features as previous works [9, 57].

**Algorithm 1** Watermarking — the learning framework.

---
1: **Inputs:** trained model $f(\cdot)$ and ID training set $S_n$;
2: $\boldsymbol{w} \sim \mathcal{N}(\mathbf{0}, \sigma_2 \mathbf{I}_d)$;                                         ▷ watermark initialization
3: **for** $t = 1$ **to** `num_step` **do**
4:     mini-batch $\{(\boldsymbol{x}_i, y_i)\}_{i=1}^m$ and noise $\{\boldsymbol{\epsilon}_i\}_{i=1}^m$;          ▷ data sampling
5:     $\mathcal{L}_m(\boldsymbol{w}) = \mathcal{L}_m^{\text{ID}}(\boldsymbol{w}) + \beta \mathcal{L}_m^{\text{OOD}}(\boldsymbol{w})$;          ▷ risk calculation
6:     $\nabla_{\boldsymbol{w}} \mathcal{L}_m(\boldsymbol{w})$ and $\boldsymbol{\kappa}$;          ▷ gradient calculation
7:     $\boldsymbol{w} \leftarrow \boldsymbol{w} - \alpha \nabla_{\boldsymbol{w}} \mathcal{L}_m(\boldsymbol{w})|_{\boldsymbol{w}+\boldsymbol{\kappa}}$;          ▷ watermark updating
8: **Output:** learned watermark $\boldsymbol{w}$.

---

Table 9: Test accuracy before/after watermarking. w/o denotes the benchmark without watermarking, SE (FE) denotes the softmax (free energy) scoring with watermarking.

| dataset | w/o | SM | FE | dataset | w/o | SM | FE |
|---------|-----|-----|-----|---------|-----|-----|-----|
| CIFAR-10 | **94.84%** | 91.85% | 93.49% | CIFAR-100 | **75.96%** | 72.03% | 74.08% |

## A.3 Backdoor Attack

Model reprogramming is also related to the *backdoor attack*, which also change models' behaviour during the test. Overall, when a backdoor is *embedded* during training and the *trigger* is activated during the test, model predictions will be modified to the *attacker-specified* labels deliberately [12]. Nowadays, *data poisoning* [12, 30, 34, 43] is among the best to realize the backdoor attack for deep models—a portion of the training sample is modified with the attacker-specified pattern (i.e., pre-defined trigger) and the attacker-specified labels. The target models are trained on these poisoned data, and the resultant models will suffer from the backdoor attack when the trigger is activated. Please refer to [29] for a comprehensive survey.

However, the backdoor attack and the model reprogramming exploit different aspects of deep models. In general, the backdoor attack utilises the *excessive* learning ability in memorizing noise features [29], while the reprogramming property states that the well-trained models can be reprogrammed for new tasks without modifying the original models.

## B  The Overall Algorithm

The overall learning framework is summarized in Algorithm 1, optimizing in a stochastic manner with `num_step` iterations. The watermark is initialized by the Gaussian noise with the $\mathbf{0}$ mean and a small standard deviation $\sigma_2 \mathbf{I}_d$ (Step 2), and the learning procedure consists of three stages for each update: (1) a set of Gaussian noise data is sampled, assuming be of the size $m$ as that of the mini-batch regarding the ID sample (Step 4); (2) the risk for ID and OOD data are computed and the overall risk is given by their sum with a trade-off parameter $\beta$ (Step 5); (3) the first-order gradient guides the pixel-level update of the watermark, using the signum of gradients and the SAM to make a reliable update (Step 7). After watermark training, the learned watermark is added to test-time inputs for OOD detection and the detection model with the pre-defined scoring function is deployed.

## C  Further Experiments

This section conducts further experiments about our proposal.

### C.1  Impact on Test Accuracy

To begin with, we study the impact of watermarking on the classification accuracy in Table 9, comparing with the results without watermarking. As we can see, watermarking has a negative impact on the test accuracy, dropping from $94.84\%$ to $91.85\%$ and $93.49\%$ on CIFAR-10; and from $75.96\%$ to $72.03\%$ and $74.08\%$ on CIFAR-100. Further, after watermarking, the classification accuracy with the free energy scoring is much better than that of the softmax scoring, with only $1.35\%$ to

Table 10: OE with/without watermarking on CIFAR benchmarks. ↓ (↑) indicates smaller (larger) values are preferred.

| | FPR95 ↓ | AUROC ↑ | AUPR ↑ |
|---|---|---|---|
| | w/ (w/o) watermark | | |
| CIFAR-10 | | | |
| iSUN | **2.70** (2.75) | **99.54** (99.55) | **99.91** (99.90) |
| Places365 | **35.05** (37.75) | **94.22** (93.12) | **98.70** (98.39) |
| Texture | **25.80** (27.95) | **95.58** (95.30) | **98.99** (98.92) |
| SVHN | **30.25** (35.85) | **95.40** (94.31) | **99.83** (99.04) |
| LSUN | 1.50 (**0.50**) | 99.71 (**99.81**) | 99.94 (99.94) |
| **average** | **19.06** (20.96) | **96.89** (96.41) | **99.39** (99.31) |
| CIFAR-100 | | | |
| iSUN | **22.85** (40.55) | **95.27** (90.60) | **98.92** (97.84) |
| Places365 | **71.75** (73.75) | **78.60** (77.89) | **94.66** (94.48) |
| Texture | **66.70** (68.20) | 81.30 (**81.50**) | 95.44 (**95.49**) |
| SVHN | 89.95 (**84.80**) | 68.20 (**72.40**) | 92.24 (**93.45**) |
| LSUN | **19.75** (34.20) | **96.17** (92.56) | **99.15** (98.32) |
| **average** | **54.20** (60.30) | **83.90** (82.99) | **96.08** (95.91) |

Table 11: MaxLogit with/without watermarking on CIFAR benchmarks. ↓ (↑) indicates smaller (larger) values are preferred.

| | FPR95 ↓ | AUROC ↑ | AUPR ↑ |
|---|---|---|---|
| | w/ (w/o) watermark | | |
| CIFAR-10 | | | |
| iSUN | **24.40** (34.90) | **96.07** (92.54) | **99.21** (98.28) |
| Places365 | 50.60 (**43.70**) | 89.03 (**89.41**) | 97.28 (**96.81**) |
| Texture | **31.35** (51.10) | **93.72** (85.88) | **98.36** (95.72) |
| SVHN | **18.80** (35.70) | **96.81** (91.25) | **99.33** (97.75) |
| LSUN | **22.65** (29.10) | **96.28** (93.73) | **99.23** (98.63) |
| **average** | **29.56** (38.90) | **94.38** (90.40) | **98.68** (97.91) |
| CIFAR-100 | | | |
| iSUN | **76.20** (78.45) | **83.18** (79.50) | **96.23** (95.06) |
| Places365 | **79.20** (80.55) | **77.19** (75.24) | **94.20** (93.42) |
| Texture | **67.75** (78.00) | **83.89** (77.15) | **96.14** (93.89) |
| SVHN | **81.10** (84.00) | **80.75** (73.66) | **95.79** (93.53) |
| LSUN | **72.60** (78.35) | **84.20** (78.95) | **96.49** (94.84) |
| **average** | **75.37** (79.87) | **81.85** (76.90) | **95.77** (94.15) |

1.88% decrease in classification accuracy. Therefore, we suggest using the free energy scoring in watermarking as a default setup, which leads to better detection capability and largely preserves the original capability in classification.

## C.2 Other Scoring Strategies with Watermarking

Note that watermarking is orthogonal to much of the existing methods and the watermarking strategy can boost many other advanced OOD detection methods. To further verify the generality and the effectiveness of our proposal, we utilize watermarking for three representative OOD detection methods, namely, OE [18], MaxLogit [16], ODIN [32], and ReAct [46].

For OE, Hendrycks et al. [18] state that the target model can benefit from fine-tuning with extra OOD training data. In general, OE requires to re-train the target model, which will be prohibitively expensive for many real-world applications. However, since the model has seen some kinds of OOD data during training, it typically reveals superior results than many other advanced detection methods.

We are interested in whether our watermarking can improve the detection capability of the models that have been fine-tuned with OE. Typically, we assume that the training-time OOD data are different from that of the test time. Therefore, following previous works [18, 33], we adopt the tiny-ImageNet [27] as the training-time OOD data for model fine-tuning. The learning objectives regarding the model parameters in OE is similar to Eq. (9) and we follow the default hyper-parameter setups as in [18]. We learn the watermarks for the fixed OE-trained models with the softmax scoring, and the results on CIFAR benchmarks are summarized in Table 10. Overall, the experimental results suggest that our watermarking can still benefit OE in effective OOD detection.

MaxLogit, ODIN, and ReAct can be viewed as the improved versions of the softmax scoring. Specifically, MaxLogit takes the maximal logit outputs in OOD scoring, which is better than softmax scoring when facing large-class setting; ODIN clamps embedding features from the second-last layer of model outputs, which can attenuate the overconfidence issue caused by the out-sized activation of abnormal hidden units; and ReAct observes that temperature scaling and adversarial feature perturbation can improve model capability in discerning OOD data from ID data.

For MaxLogit, we directly use the learning objective of the softmax scoring-based watermarking, with the corresponding scoring function of the form:

$$s_{\mathrm{MaxLogit}}(\boldsymbol{x}; f) = \max_k f_k(\boldsymbol{x}), \tag{11}$$

which directly use the logit outputs (instead of softmax outputs) in discerning ID and OOD data.

Moreover, for ODIN, the associated scoring function is given by

$$s_{\mathrm{ODIN}}(\boldsymbol{x}; f) = \max_k \mathtt{softmax}_k f(\tilde{\boldsymbol{x}})/T, \tag{12}$$

Table 12: ReAct scoring with/without water-marking on CIFAR benchmarks. ↓ (↑) indicates smaller (larger) values are preferred.

| | FPR95 ↓ | AUROC ↑ | AUPR ↑ |
|---|---|---|---|
| | w/ (w/o) watermark | | |
| CIFAR-10 | | | |
| iSUN | **27.90** (63.65) | **95.73** (87.54) | **99.13** (97.17) |
| Places365 | **62.55** (62.65) | **85.90** (86.98) | **96.50** (96.74) |
| Texture | **39.85** (58.90) | **93.68** (87.32) | **98.63** (96.86) |
| SVHN | **40.40** (43.35) | **93.47** (93.27) | **98.61** (98.60) |
| LSUN | **23.35** (59.40) | **96.29** (88.85) | **9.24** (97.51) |
| average | **38.91** (57.59) | **93.01** (88.79) | **80.42** (97.37) |
| CIFAR-100 | | | |
| iSUN | **68.05** (86.40) | **83.91** (75.31) | **96.35** (94.04) |
| Places365 | **82.65** (87.70) | **73.18** (71.20) | **93.00** (92.50) |
| Texture | **74.65** (86.35) | **78.35** (71.33) | **94.72** (92.39) |
| SVHN | **85.95** (77.50) | **75.47** (72.79) | **94.34** (92.33) |
| LSUN | **66.95** (86.85) | **84.02** (74.71) | **96.39** (93.90) |
| average | **75.65** (84.96) | **78.98** (73.06) | **94.96** (93.03) |

Table 13: ODIN scoring with/without water-marking on CIFAR benchmarks. ↓ (↑) indicates smaller (larger) values are preferred.

| | FPR95 ↓ | AUROC ↑ | AUPR ↑ |
|---|---|---|---|
| | w/ (w/o) watermark | | |
| CIFAR-10 | | | |
| iSUN | **25.05** (35.15) | **96.21** (93.09) | **99.23** (98.44) |
| Places365 | **59.60** (55.95) | **87.74** (85.84) | **97.07** (96.22) |
| Texture | **36.35** (49.50) | **93.83** (86.72) | **98.64** (96.18) |
| SVHN | **40.55** (43.20) | **93.29** (91.34) | **98.56** (97.95) |
| LSUN | **23.75** (29.40) | **96.45** (94.06) | **99.27** (98.64) |
| average | **37.06** (42.64) | **93.50** (90.21) | **98.55** (97.49) |
| CIFAR-100 | | | |
| iSUN | **69.60** (70.80) | **83.70** (81.32) | **96.32** (95.39) |
| Places365 | **82.10** (88.50) | **74.65** (72.07) | **93.64** (92.74) |
| Texture | **75.95** (82.40) | **78.80** (71.87) | **94.86** (92.50) |
| SVHN | **86.35** (74.65) | **76.04** (59.40) | **94.52** (89.21) |
| LSUN | **67.80** (71.20) | **84.61** (81.18) | **96.60** (95.47) |
| average | **76.36** (81.51) | **79.56** (73.17) | **95.18** (93.06) |

where $\tilde{\boldsymbol{x}} = \boldsymbol{x} - \xi \mathtt{sign}(-\nabla_{\boldsymbol{x}} \log \mathtt{softmax}_y f(\boldsymbol{x}))$ is the perturbed data point and $\xi$ is the perturbation magnitude. For the watermark training, the learning objectives are of the form:

$$\ell_{\mathrm{SM}}^{\mathrm{ID}}(\boldsymbol{x}, y; f) = -\log \mathtt{softmax}_y f(\tilde{\boldsymbol{x}}) \text{ and } \ell_{\mathrm{SM}}^{\mathrm{OOD}}(\boldsymbol{x}; f) = -\sum_k \frac{1}{c} \log \mathtt{softmax}_k f(\tilde{\boldsymbol{x}}), \quad (13)$$

following the same definition of $\tilde{\boldsymbol{x}}$ as in Eq. (12).

For ReAct, we assume the feature extractor defined by the second-last of model outputs by $f_{\mathrm{FEA}}$ and the above classifier by $f_{\mathrm{CLA}}$, i.e., $f(\boldsymbol{x}) = f_{\mathrm{CLA}}(f_{\mathrm{FEA}}(x))$. Then, the rectified model output is

$$f_{\mathtt{ReAct}}(\boldsymbol{x}) = f_{\mathrm{CLA}}(\min(f_{\mathrm{FEA}}(\boldsymbol{x}), \tau)), \quad (14)$$

truncating values of hidden units from the second-last layer that are above $\tau$. The corresponding learning objectives in watermark training can be written as

$$\ell_{\mathrm{SM}}^{\mathrm{ID}}(\boldsymbol{x}, y; f) = -\log \mathtt{softmax}_y f_{\mathtt{ReAct}}(\boldsymbol{x}) \text{ and } \ell_{\mathrm{SM}}^{\mathrm{OOD}}(\boldsymbol{x}; f) = -\sum_k \frac{1}{c} \log \mathtt{softmax}_k f_{\mathtt{ReAct}}(\boldsymbol{x}), \quad (15)$$

which is similar to that of the softmax scoring in Eq. (9).

Tables 11 to 13 summarize the experimental results on CIFAR-10 and CIFAR-100 datasets, where $T$ is fixed to 1000, $\xi$ is set to 0.0014, and $\tau$ is chosen such that 10% of the activation values are clamped, following the setups of the original papers. The performance improvements are illustrious regarding the performance of MaxLogit, ReAct, and ODIN with and without watermarking, largely confirming the fact that our proposed watermarking is orthogonal to existing works. Further, since we directly use the training strategy of softmax scoring-based watermarking for MaxLogit, our results in Table 11 demonstrate that watermarking can also benefit from improved choices of scoring strategies.

## C.3 Other Learning Strategies with Watermarking

Also, we consider the situation where a set of OOD data are available for watermark training, where we adopt the tiny-ImageNet [27] dataset as training-time OOD dataset. Here, we replace the Gaussian noise in the OOD learning objective $\ell^{\mathrm{OOD}}$ to be the randomly selected sample from the tiny-ImageNet dataset. Then, the learning objective with training-time OOD data is of the form

$$\mathcal{L}_n(\boldsymbol{w}) = \sum_n \ell^{\mathrm{ID}}(\boldsymbol{x}_i + \boldsymbol{w}, y_i; f) + \beta \sum_n \ell^{\mathrm{OOD}}(\boldsymbol{o}_j + \boldsymbol{w}; f), \quad (16)$$

where $\boldsymbol{o}_j$ denotes the randomly selected sample from the tiny-ImageNet dataset. The experimental results on CIFAR-10 and CIFAR-100 datasets are summarized in Table 14 and Table 15. Unfortunately,

Table 14: Softmax scoring with tiny-ImageNet and Gaussian noise on CIFAR benchmarks. ↓ (↑) indicates smaller (larger) values are preferred.

Table 15: Free energy scoring with tiny-ImageNet and Gaussian noise on CIFAR benchmarks. ↓ (↑) indicates smaller (larger) values are preferred.

| | FPR95 ↓ | AUROC ↑ | AUPR ↑ |
|---|---|---|---|
| | tiny-ImageNet (Gaussian Noise) | | |
| CIFAR-10 | | | |
| iSUN | **22.80** (43.60) | **95.88** (93.53) | **99.16** (98.67) |
| Places365 | **61.95** (60.75) | 86.47 (**87.85**) | 96.87 (**96.94**) |
| Texture | **41.45** (42.00) | **92.93** (92.83) | **98.51** (98.43) |
| SVHN | 67.15 (**27.25**) | 88.43 (**96.00**) | 97.59 (**99.17**) |
| LSUN | **22.40** (40.70) | **96.16** (94.36) | **99.22** (98.86) |
| average | 43.15 (**42.86**) | 91.97 (**92.91**) | 98.27 (**98.41**) |
| CIFAR-100 | | | |
| iSUN | **72.85** (77.85) | **81.22** (79.91) | **95.66** (95.35) |
| Places365 | **79.75** (83.25) | **75.55** (74.53) | **93.77** (93.47) |
| Texture | **73.35** (78.10) | **78.88** (77.14) | **94.61** (94.26) |
| SVHN | 83.00 (**82.95**) | 76.15 (**76.92**) | 94.52 (**94.72**) |
| LSUN | **71.65** (76.75) | **82.93** (79.60) | **96.16** (95.27) |
| average | **76.20** (76.54) | **79.10** (78.99) | **95.08** (94.96) |

| | FPR95 ↓ | AUROC ↑ | AUPR ↑ |
|---|---|---|---|
| | tiny-ImageNet (Gaussian Noise) | | |
| CIFAR-10 | | | |
| iSUN | 23.65 (**16.30**) | 95.89 (**96.97**) | 99.16 (**99.39**) |
| Places365 | **36.15** (36.25) | **91.89** (91.87) | **97.94** (97.94) |
| Texture | **31.20** (32.60) | **93.34** (93.14) | **98.39** (98.08) |
| SVHN | **13.80** (16.45) | **97.53** (97.11) | **99.49** (99.39) |
| LSUN | 23.15 (**16.85**) | 86.20 (**96.97**) | 99.20 (**99.38**) |
| average | 25.59 (**23.69**) | 92.97 (**95.21**) | **98.83** (98.83) |
| CIFAR-100 | | | |
| iSUN | **77.10** (75.05) | **83.62** (83.07) | **96.40** (96.15) |
| Places365 | **79.25** (80.50) | 77.67 (**77.78**) | 94.22 (**94.45**) |
| Texture | **68.70** (75.15) | **81.36** (79.55) | **95.10** (94.79) |
| SVHN | **82.00** (82.85) | **77.38** (75.26) | **94.80** (94.18) |
| LSUN | **71.35** (71.85) | **84.14** (84.01) | **96.55** (96.33) |
| average | **75.68** (77.08) | **80.83** (79.93) | **95.41** (95.18) |

Table 16: Softmax scoring with learning from Gaussian noise and "perm" augmentation. ↓ (↑) indicates smaller (larger) values are preferred.

Table 17: Softmax scoring with learning from Gaussian noise and "rotate" augmentation. ↓ (↑) indicates smaller (larger) values are preferred.

| | FPR95 ↓ | AUROC ↑ | AUPR ↑ |
|---|---|---|---|
| | perm (common) | | |
| CIFAR-10 | | | |
| iSUN | **38.00** (41.50) | **93.99** (93.98) | **98.79** (98.77) |
| Places365 | **55.20** (56.30) | **89.13** (89.03) | **97.49** (97.32) |
| Texture | **42.20** (43.80) | 92.30 (**93.06**) | 98.28 (**98.46**) |
| SVHN | 33.35 (**27.00**) | 94.75 (**96.07**) | 98.93 (**99.20**) |
| LSUN | **36.40** (37.85) | 94.17 (**94.57**) | 98.80 (**98.89**) |
| average | **41.03** (41.29) | 92.87 (**93.34**) | 98.46 (**98.53**) |
| CIFAR-100 | | | |
| iSUN | **77.90** (84.00) | **79.11** (75.98) | **95.11** (94.37) |
| Places365 | **79.90** (83.60) | **75.21** (73.20) | **93.68** (93.22) |
| Texture | **75.80** (83.00) | **77.13** (72.45) | **94.17** (92.78) |
| SVHN | **85.45** (87.20) | **73.22** (72.45) | **93.65** (93.45) |
| LSUN | **76.85** (81.05) | **79.37** (77.40) | **95.21** (94.75) |
| average | **79.18** (83.77) | **76.81** (74.30) | **94.36** (93.71) |

| | FPR95 ↓ | AUROC ↑ | AUPR ↑ |
|---|---|---|---|
| | rotate (common) | | |
| CIFAR-10 | | | |
| iSUN | **40.25** (41.50) | 93.44 (**93.98**) | 98.63 (**98.77**) |
| Places365 | **56.15** (56.30) | 88.12 (**89.03**) | 97.11 (**97.32**) |
| Texture | **41.15** (43.80) | 92.66 (**93.06**) | 98.34 (**98.46**) |
| SVHN | 29.65 (**27.00**) | 95.42 (**96.07**) | 99.07 (**99.20**) |
| LSUN | 37.90 (**37.85**) | 93.95 (**94.57**) | 98.76 (**98.89**) |
| average | **41.02** (41.29) | 92.72 (**93.34**) | 98.38 (**98.53**) |
| CIFAR-100 | | | |
| iSUN | **77.45** (84.00) | **78.73** (75.98) | **94.95** (94.37) |
| Places365 | **79.35** (83.60) | **75.26** (73.20) | **93.67** (93.22) |
| Texture | **76.35** (83.00) | **76.88** (72.45) | **94.16** (92.78) |
| SVHN | **84.90** (87.20) | **72.98** (72.45) | **93.58** (93.45) |
| LSUN | **78.10** (81.05) | **78.59** (77.40) | **95.00** (94.75) |
| average | **79.23** (83.77) | **76.49** (74.30) | **94.27** (93.71) |

for our current realization, we observe that watermark training with extra OOD data fails to induce a large performance improvement in OOD detection. Even worse, in some cases, learning with extra OOD data can impair the power of the resultant watermarks. We conjecture that the inductive bias introduced by the training-time OOD data may deviate from the test-time data, severely misleading the resultant watermarks in showing results even lower than that of the simple Gaussian noise.

We also list the detection performance with "perm" and "rotate" in Section 6, demonstrating the effectiveness of the resultant watermarks on far OOD cases. Here, the training objective is:

$$\mathcal{L}_n(\boldsymbol{w}) = \sum_n \ell^{\text{ID}}(\boldsymbol{x}_i + \boldsymbol{w}, y_i; f) + \beta \sum_n \ell^{\text{OOD}}(\boldsymbol{\epsilon}_j + \boldsymbol{w}; f) + \beta \sum_n \ell^{\text{OOD}}(\tilde{\boldsymbol{x}}_i + \boldsymbol{w}; f), \quad (17)$$

where $\tilde{\boldsymbol{x}}$ is an augmentation of the original $\boldsymbol{x}$ with either "perm" or "rotate". The results are summarized from Tables 16 to 19. As we can see, the results with "perm" and "rotate" is comparable with (even better than) the original learning setup with only the Gaussian noise (common).

Table 18: Free energy scoring with learning from Gaussian noise and "perm" augmentation. ↓ (↑) prefers smaller (larger) values.

|  | FPR95 ↓ | AUROC ↑ | AUPR ↑ |
|---|---|---|---|
|  | perm (common) | | |
| CIFAR-10 | | | |
| iSUN | 25.60 (**24.20**) | 95.81 (**96.10**) | 99.14 (**99.22**) |
| Places365 | 39.70 (**39.45**) | **91.96** (91.46) | **98.01** (97.89) |
| Texture | 39.15 (**38.95**) | 92.13 (**92.65**) | 97.93 (**98.16**) |
| SVHN | **16.95** (18.75) | **97.01** (96.54) | **99.37** (99.27) |
| LSUN | 22.10 (**21.80**) | **96.38** (96.27) | **99.27** (99.24) |
| **average** | 28.70 (**28.63**) | **94.66** (94.61) | 98.74 (**98.76**) |
| CIFAR-100 | | | |
| iSUN | 77.05 (**75.30**) | 83.49 (**84.51**) | 96.32 (**96.78**) |
| Places365 | 80.40 (**78.05**) | 77.11 (**78.15**) | 94.25 (**94.28**) |
| Texture | **68.85** (70.80) | **81.31** (81.14) | 95.14 (**95.02**) |
| SVHN | 81.95 (**80.50**) | **78.27** (77.27) | **95.14** (94.70) |
| LSUN | 77.20 (**75.10**) | **83.58** (83.53) | 96.35 (**97.05**) |
| **average** | 77.09 (**75.95**) | 80.75 (**81.12**) | 95.44 (**95.57**) |

Table 19: Free energy scoring with learning from Gaussian noise and "rotate" augmentation. ↓ (↑) prefers smaller (larger) values.

|  | FPR95 ↓ | AUROC ↑ | AUPR ↑ |
|---|---|---|---|
|  | rotate (common) | | |
| CIFAR-10 | | | |
| iSUN | 23.85 (**24.20**) | 96.04 (**96.10**) | 99.20 (**99.22**) |
| Places365 | **38.60** (39.45) | **91.94** (91.46) | **97.97** (97.89) |
| Texture | **35.20** (38.95) | **92.93** (92.65) | **98.26** (98.16) |
| SVHN | **16.85** (18.75) | **97.00** (96.54) | **99.37** (99.27) |
| LSUN | 22.80 (**21.80**) | 96.10 (**96.27**) | 99.19 (**99.24**) |
| **average** | **27.46** (28.63) | **94.80** (94.61) | 98.80 (**98.76**) |
| CIFAR-100 | | | |
| iSUN | 85.45 (**75.30**) | 81.39 (**84.51**) | 95.85 (**96.78**) |
| Places365 | 80.10 (**78.05**) | 76.53 (**78.15**) | 93.79 (**94.28**) |
| Texture | 71.55 (**70.80**) | 80.27 (**81.14**) | 94.67 (**95.02**) |
| SVHN | **80.20** (80.50) | **79.27** (77.27) | **95.29** (94.70) |
| LSUN | 81.25 (**75.10**) | 81.85 (**83.53**) | 95.97 (**97.05**) |
| **average** | 79.71 (**75.95**) | 79.86 (**81.12**) | 95.12 (**95.57**) |

Table 20: Comparison of watermarking and different OOD scoring functions on CIFAR benchmarks.

|  | w/o watermarking | | | | | | | w/ watermarking | | | | |
|---|---|---|---|---|---|---|---|---|---|---|---|---|
|  | Softmax [17] | Energy [33] | ReAct [46] | ODIN [32] | Mahalan-obis [28] | GradNorm [4] | OE [18] | Softmax | Energy | ReAct | ODIN | OE |
| CIFAR-10 | | | | | | | | | | | | |
| FPR95 | 55.70 | 37.67 | 57.59 | 42.64 | 34.18 | 40.51 | 20.96 | 42.86 | 23.69 | 38.91 | 37.06 | **19.06** |
| AUROC | 89.82 | 90.56 | 88.79 | 90.21 | 93.23 | 90.10 | 96.41 | 92.91 | 95.21 | 93.01 | 93.50 | **96.89** |
| AUPR | 97.32 | 97.46 | 97.37 | 97.49 | 98.41 | 97.35 | 99.39 | 98.42 | 98.83 | 80.42 | 98.55 | **99.31** |
| CIFAR-100 | | | | | | | | | | | | |
| FPR95 | 82.97 | 81.61 | 84.96 | 81.51 | 55.63 | 83.68 | 60.30 | 79.98 | 77.07 | 75.65 | 76.36 | **54.20** |
| AUROC | 73.88 | 76.50 | 73.06 | 73.17 | 82.26 | 72.93 | 82.99 | 77.57 | 79.93 | 78.98 | 79.56 | **83.90** |
| AUPR | 93.43 | 94.05 | 93.03 | 93.06 | 95.56 | 93.00 | 95.91 | 94.61 | 95.18 | 94.96 | 95.18 | **96.08** |

## C.4 Comparison with State-of-the-art Methods

For the concreteness of our discussion, this section compares our proposal with state-of-the-art methods in OOD detection. In particular, we compare with softmax scoring (Softmax) [17], free energy scoring (Energy) [33], ReAct [46], ODIN [32], Mahalanobis [28], GradNorm [21], and OE [18]. The experimental results on CIFAR benchmarks are summarized in Table 20. The average performance on iSUN, Places365, Texture, SVHN, and LSUN is reported. As we can see, watermarking can boost the performance of various scoring methods in OOD detection, achieving the best detection performance compared with all other advanced methods.

## C.5 Experiments with Mean and Standard Deviation

This section further verifies the results from Table 1 to Table 4 with five individual trails (random seeds). In Table 21, Table 22, and Table 23, we summarize the average results and the standard deviation for the softmax scoring and the free energy scoring, respectively. In Figure 5, we also depict the learned watermarks for each trial on CIFAR-10 and CIFAR-100. Overall, we observe that the learned watermarks preserve some similar pattern (e.g., the shape of areas with large values) given the same dataset and the same scoring strategy, and the improvement of watermarking is stable across different datasets and scoring methods.

## C.6 Experiments with different models

We demonstrate the effectiveness of our watermarking across various model architectures, including ResNet-18 [14], WRN-40-2, and ViT-B/16 [8]. We conduct experiments on the ImageNet benchmark and summarized the results in Table 24. As we can see, in both the softmax and free energy scoring cases, our watermarking can improve the detection performance across various models. However, the

Table 21: The softmax scoring with/without watermarking on CIFAR benchmarks. Five individual trails (mean ± std) are conducted. The notion ↓ (↑) indicates smaller (larger) values are preferred.

| | FPR95 ↓ | AUROC ↑ | AUPR ↑ |
|---|---|---|---|
| | w/ (w/o) watermark | | |
| CIFAR-10 | | | |
| iSUN | **44.68 ± 1.49** (55.43 ± 0.29) | **93.38 ± 0.28** (90.10 ± 0.22) | **97.80 ± 0.07** (97.80 ± 0.07) |
| Places365 | **59.21 ± 0.97** (60.53 ± 1.31) | **88.83 ± 0.34** (87.83 ± 0.17) | **97.08 ± 0.11** (97.06 ± 0.06) |
| Texture | **42.07 ± 1.23** (59.37 ± 1.55) | **93.03 ± 0.23** (88.56 ± 0.37) | **98.47 ± 0.07** (97.20 ± 0.09) |
| SVHN | **29.25 ± 2.17** (48.07 ± 0.97) | **95.69 ± 0.35** (91.80 ± 0.13) | **99.11 ± 0.07** (97.24 ± 0.03) |
| LSUN | **40.45 ± 2.14** (52.23 ± 1.04) | **94.11 ± 0.45** (91.50 ± 0.10) | **98.79 ± 0.11** (98.15 ± 0.04) |
| average | **43.13 ± 0.16** (55.12 ± 0.10) | **93.00 ± 0.33** (89.95 ± 0.19) | **98.25 ± 0.08** (97.49 ± 0.05) |
| CIFAR-100 | | | |
| iSUN | **78.70 ± 1.48** (82.40 ± 0.81) | **78.30 ± 0.57** (75.62 ± 0.33) | **94.92 ± 0.17** (94.11 ± 0.10) |
| Places365 | **82.55 ± 0.65** (82.97 ± 0.86) | **74.69 ± 0.50** (74.29 ± 0.29) | **93.77 ± 0.13** (93.41 ± 0.12) |
| Texture | **77.83 ± 1.47** (83.48 ± 0.70) | **76.42 ± 0.18** (73.37 ± 0.37) | **94.06 ± 0.08** (92.95 ± 0.11) |
| SVHN | **83.71 ± 2.27** (84.72 ± 0.73) | **76.16 ± 0.78** (71.29 ± 0.63) | **94.51 ± 0.17** (92.88 ± 0.22) |
| LSUN | **78.57 ± 1.09** (81.67 ± 0.78) | **78.37 ± 0.54** (75.77 ± 0.33) | **94.92 ± 0.16** (94.18 ± 0.11) |
| average | **88.27 ± 1.39** (83.04 ± 0.77) | **76.78 ± 0.51** (74.06 ± 0.39) | **94.43 ± 0.14** (93.50 ± 0.13) |

Table 22: The free energy scoring with/without watermarking on CIFAR benchmarks. Five individual trails (mean ± std) are conducted. The notion ↓ (↑) indicates smaller (larger) values are preferred.

| | FPR95 ↓ | AUROC ↑ | AUPR ↑ |
|---|---|---|---|
| | w/ (w/o) watermark | | |
| CIFAR-10 | | | |
| iSUN | **18.84 ± 2.76** (33.66 ± 0.75) | **96.28 ± 0.62** (92.62 ± 0.31) | **99.23 ± 0.13** (98.27 ± 0.10) |
| Places365 | **38.89 ± 1.74** (40.67 ± 0.91) | **91.92 ± 0.41** (89.62 ± 0.20) | **98.01 ± 0.11** (97.16 ± 0.12) |
| Texture | **34.60 ± 2.16** (52.67 ± 1.10) | **93.36 ± 0.26** (85.19 ± 0.35) | **98.31 ± 0.06** (95.40 ± 0.13) |
| SVHN | **14.96 ± 0.93** (35.60 ± 0.50) | **97.12 ± 0.15** (91.08 ± 0.22) | **99.39 ± 0.03** (97.71 ± 0.07) |
| LSUN | **16.63 ± 2.12** (27.12 ± 0.85) | **96.43 ± 0.43** (94.32 ± 0.07) | **99.26 ± 0.09** (98.70 ± 0.02) |
| average | **24.78 ± 1.94** (37.94 ± 0.82) | **95.02 ± 0.37** (90.56 ± 0.23) | **98.84 ± 0.08** (97.44 ± 0.08) |
| CIFAR-100 | | | |
| iSUN | **74.62 ± 1.97** (81.85 ± 1.14) | **84.30 ± 0.85** (78.78 ± 0.40) | **96.53 ± 0.24** (94.90 ± 0.13) |
| Places365 | **77.79 ± 0.27** (80.27 ± 1.02) | **78.13 ± 0.78** (76.46 ± 0.54) | **94.40 ± 0.31** (93.92 ± 0.16) |
| Texture | **68.96 ± 1.51** (79.47 ± 0.27) | **82.07 ± 0.62** (76.34 ± 0.34) | **95.38 ± 0.24** (93.64 ± 0.11) |
| SVHN | **80.30 ± 0.75** (85.80 ± 0.87) | **78.55 ± 0.63** (73.61 ± 0.37) | **95.11 ± 0.18** (93.51 ± 0.10) |
| LSUN | **71.25 ± 2.00** (79.26 ± 1.43) | **84.94 ± 0.54** (79.34 ± 0.40) | **96.65 ± 0.12** (94.99 ± 0.11) |
| average | **74.58 ± 1.30** (81.33 ± 0.94) | **81.59 ± 0.68** (76.90 ± 0.41) | **95.61 ± 0.21** (94.19 ± 0.12) |

improvements after watermarking on the large-scale models (i.e., ViT-B/16) are not as remarkable as that of the small models (e.g., ResNet-18). It is because that the large-scale models themselves can already excel at OOD detection (better results without watermarking than that of ResNet-18 and WRN-40-2), so there may not remain a large space for their further improvements.

### C.7   Hyper-parameter Setups

For the hyper-parameter setups in our experiments, we use random search to choose the proper $\sigma_1$ from the candidate parameter set $\{0.0, 0.2, 0.4, 0.6, 0.8, 1.0, 1.2, 1.4, 1.6, 1.8, 2.0\}$, and the proper $\rho$ from $\{0.0, 0.02, 0.05, 0.07, 0.1, 0.2, 0.5, 0.7, 1.0, 2.0, 5.0\}$. For softmax scoring, $\beta$ is chosen from $\{0.0, 0.5, 1.0, 1.5, 2.0, 2.5, 3.0, 3.5, 4.0, 4.5, 5.0\}$. For free energy scoring, $\beta$ is chosen from $\{0.0, 0.02, 0.04, 0.06, 0.08, 0.1, 0.2, 0.4, 0.6, 0.8, 1.0\}$, and $T_1, T_2$ are chosen from $\{0.1, 0.2, 0.3, 0.4, 0.5, 0.6, 0.7, 0.8, 0.9, 1.0\}$. model performance is tested on validation OOD datasets that are separated from iSUN, Places365, Texture, SVHN, and LSUN datasets.

We adopt the random search with many trials by the following three steps. Step 1: we randomly select a hyperparameter (e.g., $\beta$) and fix the values of all other hyperparameters to be their current optimal values. Step 2: we choose the best $\beta$ from the candidate set. Step 3: do Steps 1-2 again. We repeat Steps 1 and 2 for 50 times in our experiments. Further, from Tables 25 to 36, we list the performance of watermarking with different hyper-parameter settings for reference, where we fix

Table 23: Softmax and free energy scoring with/without watermarking on ImageNet. Five individual trails (mean ± std) are conducted. The notion ↓ (↑) indicates smaller (larger) values are preferred.

| | FPR95 ↓ | AUROC ↑ | AUPR ↑ |
|---|---|---|---|
| | w/ (w/o) watermark | | |
| Softmax Scoring | | | |
| iSUN | **12.69 ± 1.55** (52.38 ± 2.07) | **97.74 ± 1.23** (92.62 ± 1.31) | **99.60 ± 0.17** (98.27 ± 0.20) |
| Places365 | **70.80 ± 2.27** (73.35 ± 3.61) | **81.74 ± 1.34** (80.52 ± 0.85) | **95.32 ± 0.17** (94.88 ± 0.31) |
| Texture | **60.59 ± 2.65** (67.71 ± 2.83) | **83.46 ± 2.37** (82.47 ± 1.77) | **98.35 ± 0.06** (98.20 ± 0.17) |
| SVHN | 44.81 ± 2.03 (**28.69 ± 1.49**) | 93.72 ± 1.29 (**95.55 ± 1.13**) | 98.76 ± 0.33 (**99.14 ± 0.20**) |
| LSUN | **11.63 ± 1.47** (54.43 ± 2.15) | **97.85 ± 1.07** (91.96 ± 2.57) | **99.57 ± 0.17** (98.39 ± 0.25) |
| average | **40.10 ± 1.99** (55.31 ± 2.43) | **90.90 ± 1.46** (88.62 ± 1.52) | **98.32 ± 0.18** (97.77 ± 0.22) |
| Free Energy Scoring | | | |
| iSUN | **32.61 ± 2.21** (45.41 ± 2.84) | 93.59 ± 1.45 (**93.96 ± 1.85**) | **98.74 ± 0.20** (98.23 ± 0.28) |
| Places365 | **72.64 ± 1.37** (74.99 ± 2.50) | **79.55 ± 0.87** (78.83 ± 0.90) | **94.67 ± 0.32** (94.26 ± 0.15) |
| Texture | **67.36 ± 1.51** (67.39 ± 1.62) | **80.60 ± 1.70** (80.52 ± 1.14) | **97.00 ± 0.15** (96.90 ± 0.25) |
| SVHN | **12.92 ± 2.20** (25.85 ± 2.47) | **97.49 ± 1.00** (95.26 ± 1.51) | 99.45 ± 0.10 (**99.00 ± 0.10**) |
| LSUN | **33.53 ± 2.00** (46.68 ± 2.33) | **93.50 ± 1.50** (90.59 ± 1.70) | **98.59 ± 0.15** (97.96 ± 0.27) |
| average | **43.81 ± 1.85** (52.06 ± 2.35) | **88.94 ± 1.30** (87.83 ± 1.42) | **97.69 ± 0.18** (97.27 ± 0.21) |

Table 24: The softmax scoring and the free energy scoring with/without watermarking on the ImageNet dataset, where we adopt different models including ResNet-18, WRN-40-2, and ViT-B/16. The notion ↓ (↑) indicates smaller (larger) values are preferred.

| | Softmax Scoring | | | Free Energy Scoring | | |
|---|---|---|---|---|---|---|
| | FPR95 ↓ | AUROC ↑ | AUPR ↑ | FPR95 ↓ | AUROC ↑ | AUPR ↑ |
| ResNet-18 | **41.85** (55.60) | **90.98** (86.64) | **98.22** (97.80) | **42.87** (53.26) | **89.50** (86.42) | **97.83** (97.75) |
| WRN-40-2 | **40.50** (54.93) | **91.22** (88.57) | **98.42** (97.69) | **43.23** (52.73) | **89.10** (86.14) | **97.73** (97.15) |
| ViT-B/16 | **31.63** (34.95) | **92.52** (91.31) | **86.77** (85.13) | **20.61** (21.64) | **95.10** (94.95) | **90.87** (90.58) |

the values of all other hyper-parameters (except for the considered one) to be their optimal values. Finally, we list the results on CIFAR benchmarks regarding the free energy scoring with different values of $T$ in Table 37.

In the end, we summarize our choices of hyper-parameters, which we adopt in Section 6 for the related experiments. On CIFAR benchmarks, our method is executed for 50 epochs. The initial learning rate $\alpha = 0.01$ divided by 10 after 25 epochs. For the softmax scoring, we set $\sigma_1 = 0.4$, $\rho = 1.0$, $\beta = 3.5$ in CIFAR-10 and $\sigma_1 = 1.0$, $\rho = 0.2$, $\beta = 2.5$ in CIFAR-100; for the free energy scoring, we set $\sigma_1 = 0.6$, $\rho = 0.7$, $\beta = 0.1$, $T_1 = 0.2$, $T_2 = 0.7$ in CIFAR-10 and $\sigma_1 = 1.0$, $\rho = 0.05$, $\beta = 1.2$, $T_1 = 0.9$, $T_2 = 0.1$ in CIFAR-100. On the ImageNet benchmark, our method is executed for 10 epochs and the initial learning rate $\alpha = 0.01$ is divided by 10 after 5 epochs. We set $\rho = 0.5$, $\sigma_1 = 0.2$, $\beta = 1.5$ for the softmax scoring and $\rho = 0.05$, $\sigma_1 = 0.4$, $\beta = 0.1$ $T_1 = T_2 = 0.5$ for the free energy scoring. Further, we fix $\sigma_2 = 0.001$ and $T = 1$.

We also utilize the tuning strategy with validation OOD data that are different from the test situation, where we adopt the tiny-ImageNet here for hyper-parameter tuning. The experimental results with softmax scoring are summarized from Tables 38 to 43. As we can see, the optimal solutions chosen by tiny-ImageNet are very similar to the cases with validation sets separated from the test data, and the improvement after watermarking is remarkable as demonstrated in Table 44.

## C.8 Different Areas in the Watermark

After watermarking, the edge area of the image is overwhelmed by the watermark's pattern, while the centre part is not much affected. However, it does not mean that the centre area is not important. Instead, under the premise of maintaining the original features, the centre area also encodes useful information in OOD detection. Table 45 is a verification of this conclusion on CIFAR-10 dataset with the free energy scoring, where we employ various masks in only preserving the watermark's features with their absolute values larger (smaller) than a threshold $\chi_1$ ($\chi_2$). As we can see, even if only a small portion of the watermark is masked (e.g., $\chi_1 = 0.10$ or $\chi_2 = 10.0$), there is a large drop in performance, even lower than the results without any watermarking. It indicates that both areas of the watermark contribute, and the overall watermark works as a whole for effective OOD detection.

Table 25: Softmax scoring on CIFAR-10 with various $\sigma_1$.

| | FPR95 | AUROC | AUPR |
|---|---|---|---|
| 2.00 | 42.11 | 92.01 | 98.24 |
| 1.80 | 42.84 | 91.89 | 98.22 |
| 1.60 | 41.41 | 92.14 | 98.25 |
| 1.40 | 42.20 | 91.82 | 98.20 |
| 1.20 | 41.98 | 91.91 | 98.20 |
| 1.00 | 42.76 | 91.95 | 98.23 |
| 0.80 | 43.38 | 91.89 | 98.21 |
| 0.60 | 39.16 | 92.89 | 98.41 |
| 0.40 | **38.66** | **93.03** | **98.45** |
| 0.20 | 43.88 | 92.80 | 98.42 |
| 0.00 | 48.71 | 91.43 | 98.11 |

Table 26: Softmax scoring on CIFAR-10 with various $\rho$.

| | FPR95 | AUROC | AUPR |
|---|---|---|---|
| 5.00 | 60.02 | 87.36 | 97.15 |
| 2.00 | 46.24 | 91.61 | 98.14 |
| 1.00 | **39.12** | **92.96** | **98.42** |
| 0.70 | 41.07 | 92.77 | 98.40 |
| 0.50 | 43.55 | 92.38 | 98.34 |
| 0.20 | 42.02 | 92.68 | 98.38 |
| 0.10 | 41.99 | 92.77 | 98.41 |
| 0.07 | 42.13 | 92.79 | 98.42 |
| 0.05 | 42.06 | 92.84 | 98.42 |
| 0.02 | 44.35 | 92.34 | 98.32 |
| 0.00 | 43.04 | 92.44 | 98.32 |

Table 27: Softmax scoring on CIFAR-10 with various $\beta$.

| | FPR95 | AUROC | AUPR |
|---|---|---|---|
| 5.00 | 40.11 | 92.48 | 98.34 |
| 4.50 | 41.30 | 92.39 | 98.32 |
| 4.00 | 41.21 | 92.39 | 98.33 |
| 3.50 | **38.65** | **92.55** | **98.34** |
| 3.00 | 41.01 | 92.55 | 98.35 |
| 2.50 | 39.66 | 92.61 | 98.35 |
| 2.00 | 38.95 | 92.98 | 98.47 |
| 1.50 | 43.89 | 92.35 | 98.33 |
| 1.00 | 40.47 | 93.08 | 98.49 |
| 0.50 | 44.51 | 92.35 | 98.30 |
| 0.00 | 49.91 | 91.46 | 98.08 |

Table 28: Energy scoring on CIFAR-10 with various $\sigma_1$.

| | FPR95 | AUROC | AUPR |
|---|---|---|---|
| 2.00 | 28.49 | 94.61 | 98.74 |
| 1.80 | 28.61 | 94.56 | 98.71 |
| 1.60 | 25.36 | 95.15 | 98.85 |
| 1.40 | 28.74 | 94.85 | 98.77 |
| 1.20 | 27.48 | 94.91 | 98.78 |
| 1.00 | 26.47 | 95.02 | 98.83 |
| 0.80 | 24.99 | 95.08 | 98.85 |
| 0.60 | **24.50** | **95.29** | **98.93** |
| 0.40 | 26.40 | 94.84 | 98.78 |
| 0.20 | 27.21 | 94.73 | 98.71 |
| 0.00 | 27.97 | 94.85 | 98.75 |

Table 29: Energy scoring on CIFAR-10 with various $\rho$.

| | FPR95 | AUROC | AUPR |
|---|---|---|---|
| 5.00 | 44.20 | 91.18 | 97.92 |
| 2.00 | 29.82 | 94.32 | 98.69 |
| 1.00 | 24.56 | 95.10 | 98.86 |
| 0.70 | **24.38** | **95.20** | **98.87** |
| 0.50 | 25.96 | 95.19 | 98.85 |
| 0.20 | 25.06 | 95.32 | 98.90 |
| 0.10 | 29.21 | 94.49 | 98.69 |
| 0.07 | 28.67 | 94.78 | 98.75 |
| 0.05 | 28.72 | 94.82 | 98.81 |
| 0.02 | 27.58 | 94.93 | 98.58 |
| 0.00 | 25.38 | 94.22 | 98.78 |

Table 30: Energy scoring on CIFAR-10 with various $\beta$.

| | FPR95 | AUROC | AUPR |
|---|---|---|---|
| 1.00 | 50.71 | 91.23 | 98.10 |
| 0.80 | 42.48 | 92.42 | 98.30 |
| 0.60 | 42.57 | 92.78 | 98.42 |
| 0.40 | 33.74 | 94.07 | 98.69 |
| 0.20 | 30.12 | 94.57 | 98.74 |
| 0.10 | **23.68** | **95.35** | **98.90** |
| 0.08 | 27.08 | 94.87 | 98.76 |
| 0.06 | 25.63 | 95.15 | 98.84 |
| 0.04 | 24.58 | 95.06 | 98.79 |
| 0.02 | 25.47 | 94.86 | 98.73 |
| 0.00 | 33.31 | 92.37 | 98.08 |

Table 31: Softmax scoring on CIFAR-100 with various $\sigma_1$.

| | FPR95 | AUROC | AUPR |
|---|---|---|---|
| 2.00 | 82.37 | 76.45 | 82.37 |
| 1.80 | 79.72 | 77.25 | 94.50 |
| 1.60 | 79.50 | 77.69 | 94.57 |
| 1.40 | 79.22 | 77.64 | 94.58 |
| 1.20 | 78.28 | 78.45 | 94.74 |
| 1.00 | **76.57** | **79.06** | **94.94** |
| 0.80 | 77.88 | 78.84 | 94.91 |
| 0.60 | 76.84 | 79.13 | 94.99 |
| 0.40 | 80.47 | 76.59 | 94.31 |
| 0.20 | 81.80 | 75.87 | 94.14 |
| 0.00 | 83.07 | 73.81 | 93.48 |

Table 32: Softmax scoring on CIFAR-100 with various $\rho$.

| | FPR95 | AUROC | AUPR |
|---|---|---|---|
| 5.00 | 84.19 | 72.43 | 92.78 |
| 2.00 | 76.44 | 79.63 | 95.08 |
| 1.00 | 77.89 | 78.58 | 94.84 |
| 0.70 | 76.39 | 78.98 | 94.91 |
| 0.50 | 78.00 | 78.82 | 94.89 |
| 0.20 | **75.43** | **78.38** | **94.73** |
| 0.10 | 77.68 | 78.63 | 94.84 |
| 0.07 | 78.07 | 78.01 | 96.60 |
| 0.05 | 76.26 | 77.77 | 94.53 |
| 0.02 | 77.79 | 78.80 | 94.81 |
| 0.00 | 79.14 | 79.14 | 94.91 |

Table 33: Softmax scoring on CIFAR-100 with various $\beta$.

| | FPR95 | AUROC | AUPR |
|---|---|---|---|
| 5.00 | 78.55 | 76.57 | 94.13 |
| 4.50 | 84.54 | 72.69 | 92.99 |
| 4.00 | 79.27 | 76.01 | 93.95 |
| 3.50 | 77.56 | 76.55 | 94.11 |
| 3.00 | 79.98 | 76.33 | 94.12 |
| 2.50 | **76.43** | **77.28** | **94.29** |
| 2.00 | 79.84 | 77.23 | 94.38 |
| 1.50 | 77.38 | 78.85 | 94.89 |
| 1.00 | 76.91 | 78.45 | 94.76 |
| 0.50 | 76.75 | 79.14 | 94.97 |
| 0.00 | 83.25 | 73.73 | 93.45 |

Table 34: Energy scoring on CIFAR-100 with various $\sigma_1$.

| | FPR95 | AUROC | AUPR |
|---|---|---|---|
| 2.00 | 77.26 | 80.38 | 95.32 |
| 1.80 | 76.55 | 80.56 | 95.31 |
| 1.60 | 78.09 | 79.68 | 95.12 |
| 1.40 | 76.68 | 80.82 | 95.41 |
| 1.20 | 75.69 | 80.64 | 95.34 |
| 1.00 | **75.57** | **80.80** | **95.40** |
| 0.80 | 75.87 | 80.46 | 95.25 |
| 0.60 | 78.11 | 80.12 | 95.23 |
| 0.40 | 76.18 | 80.73 | 95.36 |
| 0.20 | 76.17 | 80.11 | 95.22 |
| 0.00 | 75.01 | 80.78 | 95.39 |

Table 35: Energy scoring on CIFAR-100 with various $\rho$.

| | FPR95 | AUROC | AUPR |
|---|---|---|---|
| 5.00 | 77.39 | 79.91 | 95.17 |
| 2.00 | 81.39 | 78.07 | 94.70 |
| 1.00 | 76.58 | 80.64 | 95.37 |
| 0.70 | 78.42 | 78.40 | 94.76 |
| 0.50 | 76.52 | 80.54 | 95.33 |
| 0.20 | 75.32 | 91.08 | 95.48 |
| 0.10 | 75.48 | 80.63 | 95.35 |
| 0.07 | 75.72 | 79.45 | 95.03 |
| 0.05 | **75.18** | **79.54** | **95.06** |
| 0.02 | 75.83 | 79.45 | 95.02 |
| 0.00 | 76.49 | 79.10 | 94.98 |

Table 36: Energy scoring on CIFAR-100 with various $\beta$.

| | FPR95 | AUROC | AUPR |
|---|---|---|---|
| 2.00 | 82.20 | 78.18 | 94.77 |
| 1.80 | 79.30 | 80.01 | 95.20 |
| 1.60 | 75.19 | 80.53 | 95.37 |
| 1.40 | 78.39 | 80.22 | 95.26 |
| 1.20 | **74.78** | **81.51** | **95.57** |
| 1.00 | 77.68 | 80.65 | 95.36 |
| 0.80 | 77.84 | 90.11 | 95.20 |
| 0.60 | 75.25 | 81.47 | 95.60 |
| 0.40 | 78.39 | 78.67 | 94.86 |
| 0.20 | 80.46 | 77.32 | 94.45 |
| 0.00 | 93.23 | 73.73 | 93.39 |

Table 37: Energy scoring on CIFAR benchmarks with various value of the hyper-parameter $T$.

| $T$ | CIFAR-10 | | | | | | | CIFAR-100 | | | | | | |
|---|---|---|---|---|---|---|---|---|---|---|---|---|---|---|
| | 1 | 5 | 10 | 50 | 100 | 500 | 1000 | 1 | 5 | 10 | 50 | 100 | 500 | 1000 |
| FPR95 | **25.9** | 27.8 | 27.7 | 28.9 | 28.4 | 28.0 | 31.2 | **74.1** | 80.4 | 77.7 | 82.3 | 80.4 | 87.0 | 89.3 |
| AUROC | **95.0** | 94.3 | 94.0 | 93.5 | 93.5 | 93.7 | 93.4 | **81.9** | 76.2 | 76.4 | 72.4 | 73.5 | 70.8 | 68.5 |
| AUPR | **98.7** | 98.5 | 98.4 | 98.2 | 98.2 | 98.2 | 98.1 | **95.7** | 94.1 | 94.1 | 92.7 | 93.0 | 91.9 | 90.9 |

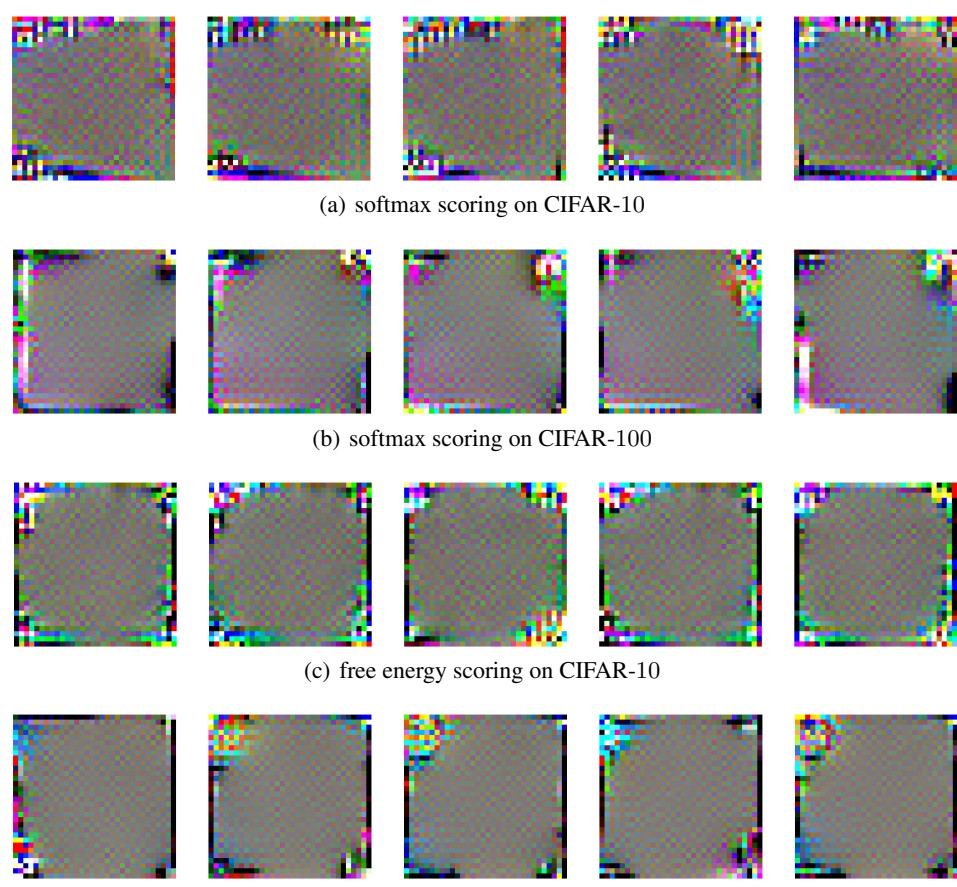

(a) softmax scoring on CIFAR-10

(b) softmax scoring on CIFAR-100

(c) free energy scoring on CIFAR-10

(d) free energy scoring on CIFAR-100

Figure 5: Illustrations of the learned watermarks with 5 individual trails.

Table 38: Softmax scoring on CIFAR-10 with various $\sigma_1$, tiny-ImageNet is adopted as the validation set.

| | FPR95 | AUROC | AUPR |
|---|---|---|---|
| 2.00 | 71.85 | 80.50 | 94.87 |
| 1.80 | 68.50 | 82.06 | 95.47 |
| 1.60 | 68.90 | 83.05 | 95.78 |
| 1.40 | 66.50 | 83.28 | 95.91 |
| 1.20 | 69.70 | 83.39 | 96.04 |
| 1.00 | 67.30 | 84.10 | 96.08 |
| 0.80 | 66.00 | 84.75 | 96.36 |
| 0.60 | 67.30 | **83.92** | 95.97 |
| 0.40 | 60.55 | 86.81 | 96.79 |
| 0.20 | 62.50 | 86.21 | 96.71 |
| 0.00 | 62.05 | 86.14 | 96.57 |

Table 39: Softmax scoring on CIFAR-10 with various $\rho$, tiny-ImageNet is adopted as the validation set.

| | FPR95 | AUROC | AUPR |
|---|---|---|---|
| 5.00 | 66.85 | 83.89 | 95.96 |
| 2.00 | 66.15 | 83.95 | 95.97 |
| 1.00 | 63.65 | 84.32 | 96.01 |
| 0.70 | 64.50 | 85.07 | 96.31 |
| 0.50 | 62.20 | 85.72 | 96.61 |
| 0.20 | 62.35 | 85.70 | 96.50 |
| 0.10 | 63.60 | 84.59 | 96.04 |
| 0.07 | 62.30 | 85.61 | 96.53 |
| 0.05 | 62.65 | 85.45 | 96.31 |
| 0.02 | 64.75 | 85.54 | 96.28 |
| 0.00 | 63.80 | 85.12 | 96.36 |

Table 40: Softmax scoring on CIFAR-10 with various $\beta$, tiny-ImageNet is adopted as the validation set.

| | FPR95 | AUROC | AUPR |
|---|---|---|---|
| 5.00 | 66.85 | 83.89 | 95.96 |
| 4.50 | 66.15 | 83.95 | 95.97 |
| 4.00 | 66.51 | 84.32 | 96.01 |
| 3.50 | 64.50 | 85.07 | 96.31 |
| 3.00 | 62.20 | 86.72 | 96.71 |
| 2.50 | 62.35 | 85.76 | 96.50 |
| 2.00 | 63.60 | 84.59 | 96.04 |
| 1.50 | 64.30 | 84.36 | 95.94 |
| 1.00 | 62.29 | 86.00 | 96.22 |
| 0.50 | 63.05 | 86.09 | 96.53 |
| 0.00 | 64.10 | 86.10 | 96.08 |

Table 41: Softmax scoring on CIFAR-100 with various $\sigma_1$, tiny-ImageNet is adopted as the validation set.

| $\sigma_1$ | FPR95 | AUROC | AUPR |
|---|---|---|---|
| 2.00 | 83.15 | 72.88 | 92.82 |
| 1.80 | 82.70 | 73.34 | 92.88 |
| 1.60 | 85.60 | 70.89 | 92.09 |
| 1.40 | 87.40 | 69.47 | 91.73 |
| 1.20 | 87.65 | 68.39 | 91.42 |
| 1.00 | 83.90 | 70.46 | 92.10 |
| 0.80 | 83.95 | 71.33 | 92.17 |
| 0.60 | 84.50 | 72.12 | 92.57 |
| 0.40 | 83.45 | 72.86 | 92.81 |
| 0.20 | **82.50** | **73.34** | **92.95** |
| 0.00 | 83.65 | 72.75 | 92.20 |

Table 42: Softmax scoring on CIFAR-100 with various $\rho$, tiny-ImageNet is adopted as the validation set.

| $\rho$ | FPR95 | AUROC | AUPR |
|---|---|---|---|
| 5.00 | 88.95 | 64.30 | 89.87 |
| 2.00 | 84.25 | 73.18 | 93.03 |
| 1.00 | **80.75** | **74.36** | **93.40** |
| 0.70 | 83.70 | 72.78 | 92.84 |
| 0.50 | 83.15 | 73.77 | 93.18 |
| 0.20 | 82.25 | 73.54 | 93.10 |
| 0.10 | 81.55 | 74.28 | 93.37 |
| 0.07 | 81.55 | 73.75 | 93.17 |
| 0.05 | 82.35 | 73.74 | 93.15 |
| 0.02 | 82.15 | 74.12 | 93.20 |
| 0.00 | 82.70 | 73.09 | 93.12 |

Table 43: Softmax scoring on CIFAR-100 with various $\beta$, tiny-ImageNet is adopted as the validation set.

| $\beta$ | FPR95 | AUROC | AUPR |
|---|---|---|---|
| 5.00 | 82.85 | 72.99 | 92.81 |
| 4.50 | 83.80 | 73.42 | 93.03 |
| 4.00 | 81.45 | 74.38 | 93.33 |
| 3.50 | **81.15** | **74.90** | **93.80** |
| 3.00 | 83.90 | 73.46 | 93.18 |
| 2.50 | 81.55 | 74.28 | 93.20 |
| 2.00 | 81.75 | 74.75 | 93.51 |
| 1.50 | 81.20 | 74.44 | 93.16 |
| 1.00 | 81.70 | 74.40 | 93.40 |
| 0.50 | 82.20 | 73.72 | 93.34 |
| 0.00 | 82.60 | 74.81 | 93.23 |

Table 44: The softmax scoring with/without watermarking on CIFAR benchmarks. Tiny-ImageNet is adopted as the validation set for hyper-parameter tuning.

| | FPR95 ↓ | AUROC ↑ | AUPR ↑ |
|---|---|---|---|
| | | w/ (w/o) watermark | |
| **CIFAR-10** | | | |
| iSUN | **29.75** (55.00) | **95.16** (89.69) | **99.00** (97.70) |
| Places365 | 65.85 (**60.10**) | 85.31 (**87.97**) | 96.49 (**97.09**) |
| Texture | **37.05** (59.60) | **93.16** (88.43) | **98.45** (97.15) |
| SVHN | **37.15** (46.70) | **93.99** (92.24) | **98.75** (98.34) |
| LSUN | **28.95** (50.75) | **95.32** (91.46) | **99.04** (98.14) |
| **average** | **39.75** (54.43) | **92.59** (89.96) | **98.35** (97.68) |
| **CIFAR-100** | | | |
| iSUN | **81.35** (82.30) | **75.90** (75.78) | **94.33** (94.15) |
| Places365 | **80.75** (82.90) | **74.50** (74.28) | **93.49** (93.21) |
| Texture | **68.30** (83.55) | **77.78** (73.30) | **94.09** (92.91) |
| SVHN | **82.60** (84.75) | **78.31** (70.64) | **95.16** (92.66) |
| LSUN | 84.15 (**81.85**) | **75.90** (74.86) | **94.35** (93.86) |
| **average** | **79.43** (83.07) | **76.47** (73.77) | **94.28** (93.35) |

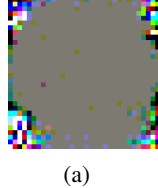

(a)

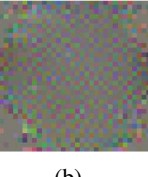

(b)

Figure 6: Masked watermarks for the free energy scoring, with (a) $\chi_1 = 1$ and (b) $\chi_2 = 1$.

Table 45: The average performance of the free energy scoring on CIFAR-10 with masking. ↓ (↑) indicates smaller (larger) values are preferred.

| | FPR95 ↓ | AUROC ↑ | AUPR ↑ |
|---|---|---|---|
| w/ watermark | **23.69** | **95.21** | **98.83** |
| w/o watermark | 37.67 | 90.56 | 97.46 |
| $\chi_1$ | | | |
| 0.10 | 30.46 | 94.41 | 98.68 |
| 1.00 | 37.72 | 93.00 | 98.52 |
| 10.0 | 37.66 | 91.87 | 97.94 |
| $\chi_2$ | | | |
| 0.10 | 42.06 | 89.88 | 97.38 |
| 1.00 | 51.28 | 87.54 | 96.86 |
| 10.0 | 36.72 | 93.08 | 98.39 |