# OpenReview forum: "Watermarking for Out-of-distribution Detection"
_NeurIPS.cc/2022/Conference — NeurIPS 2022 Accept_

### Official Review · Reviewer_qcRd · 2022-07-05

**Rating:** 4
**Confidence:** 4
**Soundness:** 1 poor
**Presentation:** 2 fair
**Contribution:** 1 poor

**Summary:**

This paper presents an out-of-distribution (OOD) detection method called “watermarking”, inspired by adversarial reprogramming. Given a scoring function that measures how in-distribution an image is, the method is to learn a fixed additive perturbation, later applied to test images, such that in-distribution images have high scores, and the perturbation has a low score. The authors claim that this will translate to being able to distinguish in vs out (i.e. OOD images with the additive perturbation will yield lower score) since OOD data was not observed during training, only the watermark. With the addition of some tricks (sign gradient descent and sharpness-aware minimization), watermarking is shown to improve OOD detection performance compared to not using watermark, on various scoring functions, and on various in-distribution datasets.

**Questions:**

- Could the authors elaborate on the motivation behind watermarking?
- What was the validation OOD dataset used to tune the hyperparameters?
- It seems like the sign is off for Equation 3 and Equation 11 (the higher the energy, the more OOD it should be). It seems like the sign is correct for the code though.
- Figure 5 seems to be missing something. The text refers to LSUN-C and LSUN-R, but there is only LSUN in Figure 5.
- The experiments testing watermarking’s impact on test accuracy (lines 272-279) doesn’t seem very relevant, since watermarking is only used to detect in vs out.
- Lines 283-284 “watermark is learned with the softmax scoring and tested regarding the free energy scoring” -> “learned with free energy and tested with softmax scoring”?
- The experiments testing the effect of masking my thresholding (lines 292-308) doesn’t seem very surprising, because the watermarks were trained to be a certain way, so there is no guarantee that changing parts of it abruptly, would do anything meaningful.


**Limitations:**

The authors have adequately addressed limitations of their work.

**Strengths And Weaknesses:**

Strengths:
- To the best of my knowledge, the idea of learning an additive perturbation to improve OOD detection is novel.
- The method is relatively simple, and doesn’t involve training a classifier or generative model.
- The method can be used with various scoring functions, and it improves the OOD detection performance most of the time.

Weaknesses:
- The method is a bit ad hoc and the motivation is unclear. The authors claim that watermarking works because “the learning procedure does not see any OOD data during training”. If the only difference between test ID and OOD is that OOD data were not seen before, then why is watermarking necessary?
- The method involves, and is sensitive to hyperparameters. Furthermore, unsurprisingly, the optimal hyperparameter setting seems to depend on the test OOD data. This limits the usage of this method. Additionally, the authors seem to have tuned the hyperparameters on a separate validation OOD dataset disjoint from the ones used to evaluate the tables, however, there is no mention of this validation dataset in the main paper of the appendix. Since the quality of the hyperparameters should depend on the validation dataset, it’s important to know how the performance changes depending on different choices of the validation datasets.
- The evaluation of the method is limited. The evaluated ID and OOD pairs are very easy to distinguish, especially since CIFAR and ImageNet are very object-centric, while the OOD datasets are mostly scenery-based (except for SVHN, which are all numbers). I would like to see at the minimum, harder problems like CIFAR-10 vs CIFAR-100, which many papers already include, or ImageNet 1k vs ImageNet 21k (excluding ImageNet 1k). It would be great to see other more difficult datasets, like the ones used in open set recognition (e.g. the semantic shift benchmark from [1]) with bigger models like the Vision Transformer.

[1] Vaze, Sagar, et al. "Open-set recognition: A good closed-set classifier is all you need."

---

> ### Author Response · Authors · 2022-08-02
> **Response to review questions (part 4)**
>
> **Revision Plan**
>
> In our revision, we will clarify our setup of hyperparameter selection in the main text, add more experiments about watermarking to verify our superiority, and move the experiments about masking into our Appendix. We will also clarify our motivation in the Introduction and fix the confusion in our draft. Thanks again for your constructive comments.
>
>
> **General Response**
>
> We have addressed your concerns about our paper. If you have more suggestions, please tell us. We will merge them into our revision as well! Since your evaluation is important for our paper, we sincerely hope that you can re-evaluate our paper if your concerns have been addressed.
>
> **References**
>
> [1] Yun-Yun Tsai, Pin-Yu Chen, and Tsung-Yi Ho. Transfer learning without knowing: Reprogramming black-box machine learning models with scarce data and limited resources. ICML (2020).
>
> [2] Gamaleldin F. Elsayed, Ian J. Goodfellow, and Jascha Sohl-Dickstein. Adversarial Reprogramming of Neural Networks. ICLR (2019).
>
> [3] Shiyu Liang, Li Yixuan, and Srikant Rayadurgam. Enhancing the reliability of out-of-distribution image detection in neural networks. NeurIPS (2017).
>
> [4] Xuefeng Du, Zhaoning Wang, Mu Cai, and Yixuan Li. VOS: Learning what you don’t know by virtual outlier synthesis. ICLR (2022).
>
> [5] Jihoon Tack, Sangwoo Mo, Jongheon Jeong, and Jinwoo Shin. CSI: Novelty detection via contrastive learning on distributionally shifted instances. NeurIPS (2020).
>
> [6] Weitang Liu, Xiaoyun Wang, John D. Owens, and Yixuan Li . Energy-based out-of-distribution detection. NeurIPS (2020).
>
> [7] Yiyou Sun, Yifei Ming, Xiaojin Zhu, and Yixuan Li. Out-of-distribution detection with deep nearest neighbors. ICML (2022).

---

> ### Author Response · Authors · 2022-08-02
> **Response to review questions (part 3)**
>
> >Q3. The evaluation of the method is limited. The evaluated ID and OOD pairs are very easy to distinguish, especially since CIFAR and ImageNet are very object-centric, while the OOD datasets are mostly scenery-based (except for SVHN, which are all numbers). I would like to see at the minimum, harder problems like CIFAR-10 vs CIFAR-100, which many papers already include, or ImageNet 1k vs ImageNet 21k (excluding ImageNet 1k). It would be great to see other more difficult datasets, like the ones used in open set recognition (e.g. the semantic shift benchmark from [1]) with bigger models like the Vision Transformer.
>
> **A3.** We have conducted extensive experiments on many benchmark datasets in the field of OOD detection (including the ImageNet OOD detection benchmark), and the results demonstrate the generality and effectiveness of our methods. However, your suggestion can further make our evaluation solid. Thus, we conduct the experiments regarding the “CIFAR10 vs. CIFAR100” setting (near-OOD detection), taking CIFAR-10 as the ID case and CIFAR-100 as OOD case.
>
> Here, we adopt the learning objective in Contrasting Shifted Instances (CSI) [5], which is well-known to be effective in near-OOD detection, with the L2 norm of feature representation (for the second last layer) as our scoring function. We follow the default hyperparameter setting in [5], further setting $\alpha=0.001$ and $\rho=0.005$ for our watermark training (since the learning objective in CSI considers both the ID and OOD cases, we do not need to specify $\beta$). The algorithm is run for 50 epochs without learning rate decay. The results are summarized as follows, which demonstrate that our watermarking is general in combination with CSI and is competent for challenging near-OOD detection tasks.
>
>
> |               | FPR95 | AUROC |  AUPR |
> |:-------------:|:-----:|:-----:|:-----:|
> | w/o watermark | 95.72 | 50.01 | 85.29 |
> |  w/ watermark | **88.95** | **65.46** | **91.44** |
>
>
> As for the experiments regarding the large-scale models, we are currently testing the effectiveness of the watermarking strategy with Vision Transformer with the ImageNet OOD detection benchmark. Since the model is a little bit large, we are still waiting for the results. We will report the results here when done!
>
> > Q4. It seems like the sign is off for Equation 3 and Equation 11 (the higher the energy, the more OOD it should be). Lines 283-284 “watermark is learned with the softmax scoring and tested regarding the free energy scoring” -> “learned with free energy and tested with softmax scoring”?
>
> **A4.** Thank you for your kind correction. We will revise the related description to eliminate the confusion.
>
> > Q5. Figure 5 seems to be missing something. The text refers to LSUN-C and LSUN-R, but there is only LSUN in Figure 5.
>
> **A5.** We mainly give the results with LSUN-R for the OOD test since we find that previous methods already perform well regarding LSUN-C. Here, for integrity, we list the OOD performance of LSUN-C with softmax scoring and free energy scoring.
>
> Softmax Scoring:
>
> | | | FPR95 | AUROC | AUPR |
> |:---------:|:-------------:|:-----:|:-----:|:-----:|
> | CIFAR-10 | w/ watermark | **18.00** | **97.94** | **99.60** |
> | | w/o watermark | 21.40 | 97.23 | 99.44 |
> | CIFAR-100 | w/ watermark | **29.35** | **96.87** | **97.00** |
> | | w/o watermark | 60.85 | 85.87 | 96.71 |
>
> Free Energy Scoring:
>
> | | | FPR95 | AUROC | AUPR |
> |:---------:|:-------------:|:-----:|:-----:|:-----:|
> | CIFAR-10 | w/ watermark | **3.85** | **99.20** | **99.84** |
> | | w/o watermark | 5.00 | 98.50 | 98.71 |
> | CIFAR-100 | w/ watermark | **14.40** | **97.45** | **99.46** |
> | | w/o watermark | 24.90 | 95.64 | 99.07 |
>
> > Q6. The experiments testing watermarking’s impact on test accuracy (lines 272-279) don’t seem very relevant since watermarking is only used to detect in vs. out.
>
> **A6.** In the original submission, we want to demonstrate that our model can excel at OOD detection while keeping the test accuracy largely intact after watermarking. Accordingly, we can do classification and detection simultaneously without feeding both the original inputs (for classification) and the watermarked inputs (for detection) into the model. It can save twice the amount of computation when deployed in real-world applications. Reporting test accuracy is also common in many other papers, such as [6,7].
>
>
> > Q7. The experiments with masking don’t seem very surprising.
>
> **A7.** In the original submission, we are curious about this experiment because the learned watermark looks like it only focuses on the edge. Your understanding is correct here. We will move more details regarding our experiments to the main content and put this part (lines 292-308) in the Appendix. Meanwhile, we can also demonstrate more experiments (e.g., CIFAR10 vs. CIFAR100) in the main content.

---

> > ### Author Response · Authors · 2022-08-05
> > **Experiments with ViT (related to Q3)**
> >
> > We use the ViT-B/16 model trained on the CIFAR-10 dataset, and the following table summarizes the average results regarding both the softmax scoring and the free energy scoring.
> >
> >
> > |                              | FPR95 | AUPR  | AUROC |
> > |:----------------------------:|:-----:|-------|:-----:|
> > | softmax w/o watermarking     | 34.95 | 91.31 | 85.13 |
> > |    softmax w/ watermarking   | **31.63** | **92.52** | **86.77** |
> > | free energy w/o watermarking | 21.24 | 94.95 | 90.58 |
> > |  free energy w/ watermarking | **20.64** | **95.10** | **90.87** |
> >
> >
> > Compared with the results given by WRN-40-2 in Table 1-2, the usage of the large-scale model can truly lead to better performance regarding both the cases with and without watermarking. However, the improvement after watermarking on the large-scale model is not as remarkable as that of the WRN-40-2 model (small-scale model). It may be due to the fact that the large-scale models themselves can already excel at OOD detection, so there does not remain a large space for their further improvements.
> >
> > We will also merge the above results into our paper. If you have more questions regarding our paper, feel free to tell us. We are very happy to discuss them with you here.

---

> > ### Author Response · Authors · 2022-08-07
> > **Further Experiments about Near OOD Detection (related to Q3)**
> >
> > For the near OOD detection in Q3, we further conduct experiments with our proposed softmax scoring-based watermarking and the free-energy scoring-based watermarking, demonstrating the power of our two realizations in Section 5 for near-OOD detection.
> >
> > In addition to the common watermarking learning setup (common) in Section 5 (ID data + random Gaussian noise), we further consider the use of shifting augmentations in CSI [5], which can be used to construct near-OOD data from the ID data. Here, we consider two representative shifting augmentations: permuting evenly partitioned data (permute) and rotating 90 degrees of original data (rotate). The shifting-augmented ID data are then taken as OOD data feed into $\ell^\text{out}(\cdot)$ along with random Gaussian noise.
> >
> >
> > The experimental results for the softmax scoring and the free-energy scoring are summarized in the following two tables. Here, the common watermark learning setup can already lead to improved performance in near-OOD detection regarding the cases without watermarking (w/o watermark). Moreover, watermarking with shifting augmentations (perm and rotate) can further boost the detection power of the models, demonstrating the effectiveness and the generality of our watermarking in near-OOD detection.
> >
> > **softmax scoring:**
> >
> > |               |   FPR95   |   AUROC   |    AUPR   |
> > |:-------------:|:---------:|:---------:|:---------:|
> > | w/o watermark |   90.10   |   55.47   |   86.16   |
> > |     common    |   88.28   |   53.16   |   84.75   |
> > |    permute    |   86.45   |   60.04   |   86.33   |
> > |     rotate    | **81.50** | **65.69** | **88.67** |
> >
> > **free-energy scoring:**
> >
> > |               |   FPR95   |   AUROC   |    AUPR   |
> > |:-------------:|:---------:|:---------:|:---------:|
> > | w/o watermark |   52.25   |   86.49   |   96.44   |
> > |     common    |   49.75   |   88.52   |   96.98   |
> > |    permute    |   48.55   |   88.40   |   97.03   |
> > |     rotate    | **47.85** | **88.90** | **97.08** |
> >
> >
> >
> > We will also merge the above results into our paper. If you have more questions regarding our paper, feel free to tell us. We are very happy to discuss them with you in the openreview system.

---

> > > ### Comment · Reviewer_qcRd · 2022-08-08
> > > **Response**
> > >
> > > Could the authors try the two new augmentations (permute and rotate) for the other experiments as well? It doesn’t seem fair or practical that the with watermark option has 3 different variants. What variant should one use in practice? Is the rotate augmentation the best overall?
> > > Also, are the results setting CIFAR10 as in or CIFAR 100? Either way, I would like to see the other variant as well.
> > >
> > > All in all, due to the heuristic nature of watermarking, I want the experiments to be very carefully done. At the current state, due to the tuning protocol, it’s hard for me to believe the results. It seems to me that watermarking is showing an improvement compared to the baseline because it gets 3 extra hyperparameters to tune, and because those parameters were tuned on the same distribution as the test set.

---

> > > > ### Author Response · Authors · 2022-08-09
> > > > **Response to your new comments**
> > > >
> > > > Thank you for your reviews. Please see our response below, or at https://openreview.net/forum?id=6rhl2k1SUGs&noteId=1mvxr7LLXCe (above your new comments).
> > > >
> > > > >Could the authors try the two new augmentations (permute and rotate) for the other experiments as well? It doesn’t seem fair or practical that the with watermark option has 3 different variants. What variant should one use in practice? Is the rotate augmentation the best overall? Also, are the results setting CIFAR10 as in or CIFAR 100? Either way, I would like to see the other variant as well.
> > > >
> > > > **Response 1:** In the above experiments, we want to reveal the power of our watermarking strategy in near OOD detection and demonstrate the possibility for further improvements.
> > > >
> > > > Following your kind suggestion, we conduct experiments on CIFAR benchmarks with the softmax scoring and the free energy scoring, and the average results (**regarding iSUN, Places365, Texture, SVHN, and LSUN**) with "perm" and "rotate" can be seen in the following table. As we can see, the results with "perm" (w/ watermark permute) and "rotate" (w/ watermark rotate) is comparable with (even better than) the original learning setup with only the Gaussian noise (w/ watermark common).
> > > >
> > > > | CIFAR-10 softmax scoring | FPR95 | AUROC |  AUPR |
> > > > |:------------------------:|:-----:|:-----:|:-----:|
> > > > |       w/o watermark      | 55.69 | 89.98 | 97.70 |
> > > > |    w/ watermark common   | 41.29 | **93.34** | **98.53** |
> > > > |   w/ watermark permute   | 41.03 | 92.87 | 98.46 |
> > > > |    w/ watermark rotate   | **41.02** | 92.72 | 98.38 |
> > > >
> > > > | CIFAR-100 softmax scoring |   FPR95   |   AUROC   |    AUPR   |
> > > > |:-------------------------:|:---------:|:---------:|:---------:|
> > > > |       w/o watermark       |   83.14   |   74.04   |   93.50   |
> > > > |    w/ watermark common    |   81.77   |   74.30   |   93.71   |
> > > > |    w/ watermark permute   | **79.18** | **76.81** | **94.36** |
> > > > |    w/ watermark rotate    |   79.23   |   76.49   |   94.27   |
> > > >
> > > > | CIFAR-10 free energy scoring |   FPR95   |   AUROC   |    AUPR   |
> > > > |:----------------------------:|:---------:|:---------:|:---------:|
> > > > |         w/o watermark        |   37.78   |   90.53   |   97.46   |
> > > > |      w/ watermark common     |   23.63   |   95.61   |   98.86   |
> > > > |     w/ watermark permute     |   23.70   |   95.66   |   98.84   |
> > > > |      w/ watermark rotate     | **22.46** | **95.80** | **98.90** |
> > > >
> > > > | CIFAR-100 free energy scoring | FPR95 | AUROC |  AUPR |
> > > > |:-----------------------------:|:-----:|:-----:|:-----:|
> > > > |         w/o watermark         | 81.41 | 76.88 | 94.13 |
> > > > |      w/ watermark common      | **76.95** | 79.72 | 95.17 |
> > > > |      w/ watermark permute     | 77.09 | **80.75** | **95.44** |
> > > > |      w/ watermark rotate      | 79.71 | 79.86 | 95.12 |
> > > >
> > > > It is difficult to make a conclusion which one is better according to our results, which is an interesting problem to study in the future.
> > > >
> > > > >All in all, due to the heuristic nature of watermarking, At the current state, due to the tuning protocol, it’s hard for me to believe the results.
> > > >
> > > > **Response 2:** We have demonstrated why the watermarking strategy works in the above new response (see https://openreview.net/forum?id=6rhl2k1SUGs&noteId=GwShy9bg7YC). We hope it can help you understand why the training objective of the watermarking strategy is useful.
> > > >
> > > > >I want the experiments to be very carefully done. It seems to me that watermarking is showing an improvement compared to the baseline because it gets 3 extra hyperparameters to tune, and because those parameters were tuned on the same distribution as the test set.
> > > >
> > > > **Response 3:** When we prepare an OOD detection work, we have to make a fair comparison with previous works. Namely, we have to use their validation sets for a fair comparison. However, we totally agree with you about hyperparameter tuning. We have conducted new experiments using tiny-ImageNet (NOT the same distribution as the test set) as a validation set. The results also show the effectiveness of our methods. We hope this can relieve your concerns a lot. In fact, using tiny-ImageNet as a validation set can also be regarded as an experimental contribution to the whole field, which is indeed more reasonable in the OOD detection problem. **Lastly, please receive our sincere thanks. Your reviews are actually helpful for the whole OOD detection community.**

---

> > > ### Author Response · Authors · 2022-08-09
> > > **Response to your new comments**
> > >
> > > Thank you for your reviews. Here are responses to your new comment below (https://openreview.net/forum?id=6rhl2k1SUGs&noteId=iMQSsvedng_).
> > >
> > > >Could the authors try the two new augmentations (permute and rotate) for the other experiments as well? It doesn’t seem fair or practical that the with watermark option has 3 different variants. What variant should one use in practice? Is the rotate augmentation the best overall? Also, are the results setting CIFAR10 as in or CIFAR 100? Either way, I would like to see the other variant as well.
> > >
> > > **Response 1:** In the above experiments, we want to reveal the power of our watermarking strategy in near OOD detection and demonstrate the possibility for further improvements.
> > >
> > > Following your kind suggestion, we conduct experiments on CIFAR benchmarks with the softmax scoring and the free energy scoring, and the average results (**regarding iSUN, Places365, Texture, SVHN, and LSUN**) with "perm" and "rotate" can be seen in the following table. As we can see, the results with "perm" (w/ watermark permute) and "rotate" (w/ watermark rotate) is comparable with (even better than) the original learning setup with only the Gaussian noise (w/ watermark common).
> > >
> > > | CIFAR-10 softmax scoring | FPR95 | AUROC |  AUPR |
> > > |:------------------------:|:-----:|:-----:|:-----:|
> > > |       w/o watermark      | 55.69 | 89.98 | 97.70 |
> > > |    w/ watermark common   | 41.29 | **93.34** | **98.53** |
> > > |   w/ watermark permute   | 41.03 | 92.87 | 98.46 |
> > > |    w/ watermark rotate   | **41.02** | 92.72 | 98.38 |
> > >
> > > | CIFAR-100 softmax scoring |   FPR95   |   AUROC   |    AUPR   |
> > > |:-------------------------:|:---------:|:---------:|:---------:|
> > > |       w/o watermark       |   83.14   |   74.04   |   93.50   |
> > > |    w/ watermark common    |   81.77   |   74.30   |   93.71   |
> > > |    w/ watermark permute   | **79.18** | **76.81** | **94.36** |
> > > |    w/ watermark rotate    |   79.23   |   76.49   |   94.27   |
> > >
> > > | CIFAR-10 free energy scoring |   FPR95   |   AUROC   |    AUPR   |
> > > |:----------------------------:|:---------:|:---------:|:---------:|
> > > |         w/o watermark        |   37.78   |   90.53   |   97.46   |
> > > |      w/ watermark common     |   23.63   |   95.61   |   98.86   |
> > > |     w/ watermark permute     |   23.70   |   95.66   |   98.84   |
> > > |      w/ watermark rotate     | **22.46** | **95.80** | **98.90** |
> > >
> > > | CIFAR-100 free energy scoring | FPR95 | AUROC |  AUPR |
> > > |:-----------------------------:|:-----:|:-----:|:-----:|
> > > |         w/o watermark         | 81.41 | 76.88 | 94.13 |
> > > |      w/ watermark common      | **76.95** | 79.72 | 95.17 |
> > > |      w/ watermark permute     | 77.09 | **80.75** | **95.44** |
> > > |      w/ watermark rotate      | 79.71 | 79.86 | 95.12 |
> > >
> > > It is difficult to make a conclusion which one is better according to our results, which is an interesting problem to study in the future.
> > >
> > > >All in all, due to the heuristic nature of watermarking, At the current state, due to the tuning protocol, it’s hard for me to believe the results.
> > >
> > > **Response 2:** We have demonstrated why the watermarking strategy works in the above new response (see https://openreview.net/forum?id=6rhl2k1SUGs&noteId=GwShy9bg7YC). We hope it can help you understand why the training objective of the watermarking strategy is useful.
> > >
> > > >I want the experiments to be very carefully done. It seems to me that watermarking is showing an improvement compared to the baseline because it gets 3 extra hyperparameters to tune, and because those parameters were tuned on the same distribution as the test set.
> > >
> > > **Response 3:** When we prepare an OOD detection work, we have to make a fair comparison with previous works. Namely, we have to use their validation sets for a fair comparison. However, we totally agree with you about hyperparameter tuning. We have conducted new experiments using tiny-ImageNet (NOT the same distribution as the test set) as a validation set. The results also show the effectiveness of our methods. We hope this can relieve your concerns a lot. In fact, using tiny-ImageNet as a validation set can also be regarded as an experimental contribution to the whole field, which is indeed more reasonable in the OOD detection problem. **Lastly, please receive our sincere thanks. Your reviews are actually helpful for the whole OOD detection community.**

---

> ### Author Response · Authors · 2022-08-02
> **Response to review questions (part 2)**
>
> > Q1.3. The authors claim that watermarking works because “the learning procedure does not see any OOD data during training.” If the only difference between test ID and OOD is that OOD data were not seen before, then why is watermarking necessary?
>
> **A1.3.** There is a misunderstanding in this comment. The watermarking strategy works due to the reprogramming property of DNNs (please see A2). The sentence “the learning procedure does not see any OOD data during training” appears when we design the learning procedure regarding watermarks (more like technical details instead of the principle that watermarks can work on OOD detection). This sentence itself does not imply the general reason why watermark works.
>
> As for the design of this learning procedure, our aim is to train the watermark to recognize the watermark with/without ID pattern (with low/high confident predictions). After training the watermark, the ID classifier will make low confident predictions for those watermarked OOD data since the patterns of OOD data largely deviate from the ID cases. Note that, the definition of OOD data is the data whose label set is disjoint with the label set of ID data, meaning that OOD data are not seen before.
>
> We further clarify the necessity of watermarking strategy. Given a fixed ID classifier, previous studies design many scoring functions to help identify OOD data. However, previous scoring strategies cannot fully distinguish ID and OOD cases and can hardly make adaptations for specified tasks and datasets. Thus, the necessity of watermarking strategy lies in the fact that it can largely boost the OOD detection capability of ID classifiers for considered tasks with a learned watermark (a static pattern) added to test-time inputs. **The effectiveness of the watermarking strategy is supported by model reprogramming [1,2] (in principle), verified by our extensive experiments (in practice), and supported by the other two reviewers.**
>
> >Q2. What was the validation OOD dataset used to tune the hyperparameters? The method involves, and is sensitive to hyperparameters. Furthermore, unsurprisingly, the optimal hyperparameter setting seems to depend on the test OOD data. This limits the usage of this method. Additionally, the authors seem to have tuned the hyperparameters on a separate validation OOD dataset disjoint from the ones used to evaluate the tables, however, there is no mention of this validation dataset in the main paper of the appendix. Since the quality of the hyperparameters should depend on the validation dataset, it’s important to know how the performance changes depending on different choices of the validation datasets.
>
> **A2.** There exists a set of validation datasets that are separated from the test datasets, following the setups such as [3,4] for a fair comparison. The detailed description of hyperparameter tuning can be found in Appendix C.6, and it is also summarized below for your reference.
>
> We adopt validation OOD datasets that are separated from the original iSUN, Places365, Texture, SVHN, and LSUN. Further, we choose the proper $\sigma_1$ from the candidate parameter set {0.0,0.2,0.4,0.6,0.8,1.0,1.2,1.4,1.6,1.8,2.0}, and the proper $\rho$ from {0.0,0.02,0.05,0.07,0.1,0.2,0.5,0.7,1.0,2.0,5.0}. For softmax scoring, $\beta$ is chosen from {0.0,0.5,1.0,1.5,2.0,2.5,3.0,3.5,4.0,4.5,5.0}; and for free energy scoring, $\beta$ is chosen from {0.0,0.02,0.04,0.06,0.08,0.1,0.2,0.4,0.6,0.8,1.0}. We will move the hyperparameter selection part to the main content in our revision.
>
> Moreover, we have shown how the hyperparameters affects the performance of our methods in Tables 17 - 28 (the 12 tables in page 20 in the original submission).
>
> We also demonstrate the detailed performance of our methods on different validation datasets. The results (regarding FPR95) with softmax scoring on CIFAR-10 dataset can be found below, where we select a set of candidate hyperparameter setups. As we can see, our selected hyperparameters (last line) is preferred across all the considered datasets.
>
>
> | $\sigma_1$ | $\rho$ | $\beta$ | iSUN  | Places365 | Texture | SVHN  | LSUN  |
> |:------------:|:--------:|:---------:|:-------:|:-----------:|:---------:|:-------:|:-------:|
> | 2.0        | 5.0    | 5.0     | 92.50 |   92.75   | 97.80   | 99.10 | 92.55 |
> | 1.6        | 1.0    | 4.0     | 50.15 |   77.85   | 58.40   | 76.65 | 47.50 |
> | 1.2        | 0.5    | 3.0     | 43.90 |   75.75   | 58.10   | 89.15 | 50.70 |
> | 0.8        | 0.1    | 2.0     | 44.75 |   66.20   | 40.80   | 37.00 | 45.00 |
> | 0.4        | 0.05   | 1.0     | 50.50 |   67.00   | 45.05   | 35.15 | 45.60 |
> | 0.4        | 1.0    | 3.5     | **40.60** | **61.15**     | **40.15**   | **29.85** | **40.40** |

---

> > ### Comment · Reviewer_qcRd · 2022-08-08
> > **Response**
> >
> > **Response to A1.3**
> >
> > The sentence that I’m referring to was used in line 132-133, which to my understanding is explaining why watermarking works:
> > “From the lens of our model, the scores should remain low if a watermarked OOD input is given since the learning procedure does not see any OOD data during training, and only the watermark can be observed”.
> >
> > Also, I’m not sure I am convinced about this statement:
> > “After training the watermark, the ID classifier will make low confident predictions for those watermarked OOD data since the patterns of OOD data largely deviate from the ID cases”. It is true that OOD data has a disjoint label set than ID data. However, the very problem of OOD detection is that our networks are easily fooled by the very OOD data that largely deviates from the ID classes. It’s not convincing to me that a normal classifier gets fooled by OOD data, but watermarking does not.
> >
> > Lastly, as I mentioned in my response to A1.1 + A1.2, model reprogramming has access to target domain data, but watermarking does not. So it doesn’t make sense to me that watermarking can adapt for specified tasks and datasets and therefore, watermarking doesn’t add anything more principled compared to the previous scoring methods.
> >
> > **Response to A2**
> >
> > Am I understanding correctly that the hyperparameters were tuned on the validation sets corresponding to  the test OOD datasets? If so, I think it’s wrong to tune hyperparameters with data that are from the same distribution as the test OOD data. Tuning the hyperparameters this way gives the model access to the very OOD classes that the model is trying to distinguish, and this is an extra degree of freedom that the baseline does not get to enjoy. The correct way to tune the hyperparameters would be to use a validation dataset that is semantically separate from the test OOD datasets, for example in [1] (see appendix A). Ideally, the baseline (no watermark) hyperparameters (learning rate at the very least) should also be tuned on the validation dataset such that both methods are calibrated on the same dataset. This is reasonable, since models are usually tuned on the in-distribution validation set– we can additionally tune for OOD detection performance on a separate OOD validation dataset.
> >
> > After reading author responses to all the reviews, I have further questions. First of all, how exactly were the hyperparameters tuned? It couldn’t have been grid search as there are too many possibilities. In tables 17-28, what were the values of the other parameters that were not varied? In my opinion, the best way to tune in such a setting would be to use random search [2] with many trials (20 for example). It seems like the authors have 6 random trials shown in their response to reviewer dM3K. I would be happy to see an extension of that table such that:
> > - The baseline (without watermark) hyperparameters are tuned on tiny ImageNet (or any other validation set that doesn’t overlap semantically with the test OOD datasets).
> > - The proposed idea is tuned with random search with more than 6 trials on the same validation dataset as above.
> >
> > Also, it seems like table 4 for $\rho$ seems to be the same as Table 19, which for $\beta$. Depending on which table is true, lines 264-271 should be corrected.
> >
> > Lastly, the “Ablation study” section should be renamed, as studying the effect of the hyperparameters is not technically an ablation.
> >
> > [1] Hendrycks, Dan, Mantas Mazeika, and Thomas Dietterich. "Deep anomaly detection with outlier exposure." arXiv preprint arXiv:1812.04606 (2018).
> >
> > [2] Bergstra, James, and Yoshua Bengio. "Random search for hyper-parameter optimization." Journal of machine learning research 13.2 (2012).

---

> > > ### Author Response · Authors · 2022-08-09
> > > **Response to your new comments (part 2)**
> > >
> > > >**Subquestion 3**: "First of all, how exactly were the hyperparameters tuned? It couldn’t have been grid search as there are too many possibilities. In tables 17-28, what were the values of the other parameters that were not varied? "
> > >
> > > **Response 3**: As your guess, we adopt the random search with many trials. Step 1: we randomly select a hyperparameter (e.g., $\beta$) and fix the values of all other hyperparameters to be their optimal values. Step 2: we select the best $\beta$ from the set. Step 3: do Steps 1-2 again. We repeat Steps 1 and 2 50 times in our experiments. We will further emphasize our hyper-parameter tuning strategy in our revision (e.g., adopt more advanced discrete optimization methods).
> > >
> > > >**Subquestion 4**: "In my opinion, the best way to tune in such a setting would be to use random search [2] with many trials (20 for example)."
> > >
> > > **Response 4**: We totally agree with you. We will follow your kind suggestions of using random search in [2] (or a more advanced method, expecting your suggestions) and tiny-ImageNet in replacement of our current tuning strategy. When we get results, we will report them here. However, due to the time limitation, we cannot demonstrate them in the uploaded revision.
> > >
> > > >**Subquestion 5**: "The baseline (without watermark) hyperparameters are tuned on tiny ImageNet (or any other validation set that doesn’t overlap semantically with the test OOD datasets)."
> > >
> > > **Response 5**: In our main content, the hyperparameter $T$ is the only tunable value for the baseline methods. $T$ is taken from the set $\\{1, 5, 10, 50, 100, 500, 1000\\}$. The following tables show the results with tiny-ImageNet being the OOD dataset (CIFAR-10/100 being the ID dataset) with different values of $T$. As shown in the following table, $T=1$ (as suggested in previous papers) is the best in general.
> > >
> > > | CIFAR-10 |     1     |     5     |   10  | 50    | 100   | 500   | 1000  |
> > > |:--------:|:---------:|:---------:|:-----:|-------|-------|-------|-------|
> > > |   FPR95  |   33.45   | **31.75** | 34.70 | 37.80 | 38.00 | 39.25 | 35.60 |
> > > |   AUROC  | **92.69** |   92.47   | 91.45 | 90.12 | 90.10 | 89.37 | 90.38 |
> > > |   AUPR   | **98.25** |   98.17   | 97.80 | 97.48 | 97.47 | 97.17 | 97.40 |
> > >
> > > | CIFAR-100 |     1     |   5   |   10  | 50    | 100   | 500   | 1000  |
> > > |:---------:|:---------:|:-----:|:-----:|-------|-------|-------|-------|
> > > |   FPR95   | **59.55** | 59.75 | 61.90 | 66.95 | 68.35 | 94.40 | 69.65 |
> > > |   AUROC   | **85.10** | 83.91 | 82.48 | 79.84 | 80.53 | 79.52 | 79.01 |
> > > |    AUPR   | **96.25** | 95.92 | 95.54 | 94.54 | 94.95 | 94.40 | 93.65 |
> > >
> > > >**Subquestion 6**: The proposed idea is tuned with random search with more than 6 trials on the same validation dataset as above.
> > >
> > > **Response 6**: The following table lists the results on CIFAR-10 with softmax scoring and tiny-ImageNet as an OOD validation dataset with 6 individual trials. We will conduct more experiments with random search regarding tiny-ImageNet and provide the results in our revision.
> > >
> > > | $\sigma_1$ | $\rho$ | $\beta$ | FPR95 | AUROC |  AUPR |
> > > |:----------:|:------:|:-------:|:-----:|:-----:|:-----:|
> > > |     0.8    |   2.0  |   1.0   | 58.45 | 87.62 | 96.51 |
> > > |     0.6    |   0.7  |   2.5   | 65.25 | 84.04 | 95.89 |
> > > |     1.6    |   0.2  |   2.5   | 68.70 | 83.56 | 95.97 |
> > > |     1.2    |  0.07  |   0.5   | 66.35 | 83.74 | 95.82 |
> > > |     0.4    |   1.0  |   4.0   | 64.30 | 95.29 | 96.25 |
> > > |     4.0    |   5.0  |   3.0   | 87.90 | 68.93 | 91.81 |
> > >
> > > >**Subquestion 7**: "It seems like table 4 for seems to be the same as Table 19. Depending on which table is true, lines 264-271 should be corrected."
> > >
> > > **Response 7**: Many thanks for your kind correction. We will update all the tables that lead to the confusion.
> > >
> > > >**Subquestion 8**: "Lastly, the “Ablation study” section should be renamed, as studying the effect of the hyperparameters is not technically an ablation."
> > >
> > > **Response 8**: Sincerely thank you for your suggestions, we will rename this section about hyperparameter selection.

---

> > > ### Author Response · Authors · 2022-08-09
> > > **Response to your new comments (part 1)**
> > >
> > > Thank you for your comments. Note that a reply to your "Response to A1.3" can be found at https://openreview.net/forum?id=6rhl2k1SUGs&noteId=GwShy9bg7YC (another reply to your new comments). We will focus on your "Response to A2" here.
> > >
> > > >**Subquestion 1**: "Am I understanding correctly that the hyperparameters were tuned on the validation sets corresponding to the test OOD datasets? If so, I think it’s wrong to tune hyperparameters with data that are from the same distribution as the test OOD data. Tuning the hyperparameters this way gives the model access to the very OOD classes that the model is trying to distinguish, and this is an extra degree of freedom that the baseline does not get to enjoy. The correct way to tune the hyperparameters would be to use a validation dataset that is semantically separate from the test OOD datasets".
> > >
> > > **Response 1**: We totally agree with you. For fair comparison, we follow [3, 4]: we do not use any data point in considered test datasets, and the adopted validation datasets follow the setups in many previous works [3,4]. It means that our hyper-parameter tuning strategy is proper in comparison with previous works, and the training procedure does not involve any data about test OOD cases.
> > >
> > > However, your suggestion of using a semantically different validation dataset is correct (we totally agree with this point), and we believe it will become the standard tuning strategy in the future in this field. Therefore, we will completely follow your suggestion with tiny-ImageNet for hyperparameter tuning in our revision. Here, we want to demonstrate that our watermarking strategy is robust to various hyper-parameter settings, due to our observations that there exists a similar trend in preference of hyperparameters, in **using the average detection performance on adopted validation datasets (current strategy)** and **using the detection performance on tiny-ImageNet**.
> > >
> > > The following results regarding FPR95 (softmax scoring-based watermarking on CIFAR-10) is a verification of our claim, where the "**candidate**" rows represent randomly selected sets of hyperparameter setups from Appendix C.6. We find that for your suggested tuning strategy with tiny-ImageNet, the optimal one (the last row "**optimal**" in the following table) is the same as that used in our paper. We sincerely appreciate your constructive suggestions in hyper-parameter tuning, and we will follow your suggestion with tiny-ImageNet in our revision, replacing all the results from Table 17-28.
> > >
> > > |           | $\sigma_1$ | $\rho$ | $\beta$ | Average Validation | tiny-ImageNet |
> > > |:---------:|:----------:|:------:|:-------:|:------------------:|:-------------:|
> > > | candidate |     2.0    |   5.0  |   5.0   |        94.94       |     95.05     |
> > > | candidate |     1.6    |   1.0  |   4.0   |        62.11       |     74.50     |
> > > | candidate |     1.2    |   0.5  |   3.0   |        63.52       |     72.70     |
> > > | candidate |     0.8    |   0.1  |   2.0   |        46.75       |     62.50     |
> > > | candidate |     0.4    |  0.05  |   1.0   |        48.66       |     62.90     |
> > > |  optimal  |     0.4    |   1.0  |   3.5   |      **42.43**     |   **58.30**  |
> > >
> > > >**Subquestion 2**: "Ideally, the baseline (no watermark) hyperparameters (learning rate at the very least) should also be tuned on the validation dataset such that both methods are calibrated on the same dataset. This is reasonable, since models are usually tuned on the in-distribution validation set– we can additionally tune for OOD detection performance on a separate OOD validation dataset."
> > >
> > > **Response 2**: In fact, many previous works directly use well-trained ID classifiers in discerning ID and OOD data, and there are not any training procedures that involve learning rate tuning (i.e., no hyperparameters actually). For other hyperparameters, such as $T$ in free energy scoring, we have adopted the suggested settings in their original paper that lead to the optimal solutions (**we also tune it according to your comment, please see results in the Response 5**). Therefore, the comparison is fair. If we miss something, we are happy to discuss and add experiments if needed.

---

> ### Author Response · Authors · 2022-08-02
> **Response to review questions (part 1)**
>
> Many thanks for your constructive comments and kind suggestions! Please find our responses below.
>
> > Q1.1. The motivation is unclear. Could the authors elaborate on the motivation behind watermarking?
>
> **A1.1.** We are motivated by model reprogramming [1,2], stating that the model can be repurposed for new tasks without modifying its parameters. It facilitates our watermarking, aiming to boost the performance of previous scoring strategies that may produce unsatisfactory results in many cases (cf., Fig. 2). Here is a detailed description for your reference, which is a brief summary of our Introduction.
>
> In previous works, post-hoc OOD detection largely relies on scoring functions built upon well-trained ID classifiers. However, one can hardly adjust these advanced methods for specific tasks since we prefer models with intact parameters. Therefore, how to boost the performance of existing scoring-based OOD detection is a very attractive problem. In this paper, we suggest that one can learn a watermark added to test-time inputs, of which pattern can be learned for specific scoring strategies and datasets. Our method is well supported by previous studies in model reprogramming [1,2], which states that **a model (with its fixed parameters) can be repurposed for a new task by modifying the pattern of inputs**. Thus, we want to reprogram the fixed ID classifier to fit a new task: identifying OOD data.
>
> Also, our work is very different from previous works in studying reprogramming properties of deep models, and their padding strategies [1] will destroy models’ original capability in OOD detection (also agreed by Reviewer WTn8). We overcome these drawbacks, and thus the watermarking performs well in OOD detection.
>
> Overall, it is the first time that model reprogramming is studied in the literature on OOD detection, and our proposal provides a new road that can motivate more works in this area. This fact is also recognized by the other two reviewers. We will continue elaborating on the motivation behind watermarking in our revision.
>
> > Q1.2. The method is a bit ad-hoc.
>
> **A1.2.** Our methods are not designed for specific datasets. They are general methods and can be used in more scenarios, which can be verified from the following two perspectives.
>
> From the principle perspective, the effectiveness of our methods takes root in the reprogramming property of DNNs. This property of DNNs has been verified in many academic papers [1,2], ranging from image classification to time-series analysis. It supports that we can **reprogram a deep model to complete other tasks**. Thus, our watermarking strategy, motivated by the reprogramming property, **is capable of reprogramming the ID classifier to complete OOD detection tasks**.
>
> From the experimental perspective, we have tested the performance of our methods across a set of datasets (cf., Section 6 and Appendix C.5) and scoring functions (cf., Appendix C.1-C.2). Therein, our watermarking can achieve better performance across all the considered settings (e.g., various datasets, scoring functions, and learning strategies), **further verifying the generality of our watermarking in the area of OOD detection**.

---

> > ### Comment · Reviewer_qcRd · 2022-08-08
> > **Response**
> >
> > I’m still not sure how model reprogramming (idea that a model can be repurposed for a new task by modifying the pattern of the inputs) will necessarily lead to improved OOD detection performance. The only thing that watermarking guarantees is what it is trained for– higher score for ID + watermark, and low score if only watermark + some perturbation is observed. There is nothing in this setup that makes it such that unknown input + watermark will result in lower score.
> > Furthermore, model reprogramming has access to the target dataset, which explains why it works on the target domain. However, in OOD detection we never have access to the test OOD dataset, and by definition we are dealing with unknown unknowns. Therefore, it is hard for me to believe that the reprogramming property has anything to do with explaining how watermarking is supposed to help with OOD detection. Please let me know if I’m missing something.

---

> > > ### Author Response · Authors · 2022-08-09
> > > **Response to your new comments**
> > >
> > > **Principle-level response to your concern:**
> > >
> > > First of all, we want to thank you. This is very good critical thinking regarding our paper. Sorry for the previous misunderstandings. Now, we realize your major concerns. We think the following sentence might address your concerns. **In the watermarking strategy, we want to reprogram previous ID classifiers to help increase the classifiers' confidence in ID data and decrease the corresponding confidence when the classifiers do not see any ID pattern.** Since the reprogrammed models are more confident in ID classes, we can expect to relieve the issue of overconfidence in ID classifiers. Note that, the overconfidence issue of ID classifiers is the main reason why ID classifiers will recognize the OOD data as ID data. If you think this can help you address your concerns, we will update them in our revision. If not, welcome more thoughts!
> > >
> > > Then, we want to mention that many OOD detection methods cannot see the OOD data in advance, yet they can still improve the OOD detection performance. The main reason is that they focus on enhancing the confidence of ID classifiers on ID data (one simple but representative way is to use the temperature functions). The related theory is still missing why we can perform OOD detection by only using the ID data (we cannot see OOD data yet we can distinguish between ID and OOD data). One possible reason is that the ID classifier might be regarded as a good one-class classifier (ID classes as the only class).
> > >
> > > **More details:**
> > >
> > > Given the main aim of the watermarking strategy (given above): we want to reprogram previous ID classifiers to help increase the classifiers' confidence in ID data. Overall, making the model have low confidence for random noise (effect of $\ell^\text{out}(\cdot)$) can play the role of regularization, such that the training procedure will not find a trivial solution that always returns a high confident prediction for any watermarked data point (effect of $\ell^\text{in}(\cdot)$).
> > >
> > > Then, since the watermark is trained such that the model will **only** produce a high confident prediction for the watermarked ID cases, it is proper to assume that the model can return low confidence for those unseen watermarked OOD data. From another lens,  the key issue in OOD detection is that the model can be overconfident in unseen data. Therefore, we make the scores of ID data higher, such that we can better distinguish between ID and OOD cases.
> > >
> > > Our above explanations are verified by our extensive experiments. However, we sincerely appreciate your concerns and we believe that future exploration can lead to many improved learning methods for watermarking, which requires our further studies.

---

> ### Author Response · Authors · 2022-08-06
> **Looking forward to your responses or further suggestions/comments!**
>
> Dear Reviewer qcRd,
>
> We have completed the experiments regarding large-scale models using ViT, please see details in https://openreview.net/forum?id=6rhl2k1SUGs&noteId=Mtd4mcWvqrZ .
>
> Now, we have addressed your initial concerns regarding our paper and provided the required experimental results. We are happy to discuss them with you in the openreview system if you still have some concerns/questions. We also welcome new suggestions/comments from you!
>
> Best regards,
>
> Authors of #1621

---

> > ### Author Response · Authors · 2022-08-08
> > **Further Discussion**
> >
> > Dear Reviewer qcRd:
> >
> > Thanks for your great efforts in reviewing and good questions here. We really hope that our answer can help to clarify. Since the discussion due is approaching, please let us know if anything we could further clarify.
> >
> > Best regards,
> >
> > Authors of #1621

---

### Official Review · Reviewer_WTn8 · 2022-07-10

**Rating:** 8
**Confidence:** 5
**Soundness:** 4 excellent
**Presentation:** 4 excellent
**Contribution:** 4 excellent

**Summary:**

OOD detection aims to detect OOD data given only ID data or ID classifier. Since OOD data may cause risks when deploying the deep learning models into the real world, it is important to detect OOD data (like this paper considering). Based on my experience, this paper considers a setting where only ID classifiers are available and it is unavailable to see the ID data. Thus, the performance on ID classes will be maintained naturally. Given only the models, previous methods proposed score functions to determine if a data point is OOD, which is a natural way to utilize the models’ information. However, this paper argues that there might be another way to use the models, i.e., reprograming the models via changing data slightly. Compared to previous methods, this paper considers using the models from a higher level (based on the model reprograming property of deep models), which is very novel and interesting.

From the technical level, the proposed watermarking strategy is also novel and different from previous reprograming methods. Extensive experiments verify that the proposed OOD detection methods are useful and performs better in general.

**Questions:**

Please see weakness.

**Limitations:**

The authors have addressed the limitation part. There are no ethical concerns regarding the proposed strategy.

**Strengths And Weaknesses:**

Pros:

1.  Reprogramming property of deep models is first used for OOD detection, which makes good contributions to the field of OOD detection. Previous methods only use models’ outputs and do not fully use the reprogramming property of deep models. This paper considers this property and successfully uses this property to address the OOD detection problem, which is novel.

2. In previous post-hoc OOD detection methods, there are no tuning parameters after selecting a score function, which does not fully use the models’ information. However, this paper breaks through this situation and shows that we can further utilize the models’ information.

3. Previous reprogramming methods cannot be directly used for OOD detection, and this paper makes a new contribution to this field by proposing the watermarking strategy, which is novel.

4. Experiments verify the performance of the watermarking strategy, which is solid evidence that the reprogramming property of deep models can also help address the OOD detection problem. This finding will motivate more work in this field. It is appreciated that the performance is also tested on ImageNet benchmark datasets.

Cons:

1. When training the watermarks, additional noise is introduced. The proposed methods seem to use more information than previous methods. Why do previous methods not use this information? Are your methods the only methods that can use this information in post-hoc OOD detection? More explanations are required here.

2. It is better to explain the use of Eq. (6). This optimization procedure seems very important to the method but lacks enough explanation.

3. In the free-energy version, how to set the hyperparameter T? How does the value of T influence the performance of free-energy watermarking method?

4. The results regarding ImageNet benchmark should be moved into the main context instead of the appendix, which can help demonstrate the performance of watermarking strategy better.

5. In the whole paper, the data use ID and OOD to represent the ID or OOD data. While the loss functions use IN and OUT to present parts regarding ID and OOD. Keeping both consistencies is a better option.

6. Adding some specific noise into data can have a great impact on the model’s output, which has been verified in many areas. Watermark is one of them. How to analyse the power of watermark in math? Necessary discussions are required.

---

> ### Author Response · Authors · 2022-08-02
> **Response to review questions (part 2)**
>
> **References**
>
>
> [1] Qizhou Wang, Feng Liu, Bo Han, Tongliang Liu, Chen Gong, Gang Niu, Mingyuan Zhou, and Masashi Sugiyama. Probabilistic margins for instance reweighting in adversarial training. NeurIPS (2021).
>
> [2] Aleksander Madry, Aleksandar Makelov, Ludwig Schmidt, Dimitris Tsipras, and Adrian Vladu. Towards Deep Learning Models Resistant to Adversarial Attacks. ICLR (2018).
>
> [3] Weitang Liu, Xiaoyun Wang, John D. Owens, and Yixuan Li . Energy-based Out-of-distribution Detection. NeurIPS (2020).

---

> ### Author Response · Authors · 2022-08-02
> **Response to review questions (part 1)**
>
> We sincerely thank you for your constructive comments and kind suggestions! Please find our responses below.
>
> > Q1. Why do previous methods not use the Gaussian Noise? Are your methods the only methods that can use this information in post-hoc OOD detection?
>
> A1. Your understanding is correct. Our method is the only one in post-hoc OOD detection that can use this information. Previously, researchers mainly focus on devising various scoring functions, which cannot make effective adaptation for tasks or other information.
> The capability in using other information reflects our flexibility in effective detection, even with such weak knowledge as Gaussian noise (without any cost to obtain). In Appendix C.1-C.2 (cf., Supplementary Material), we further demonstrate that stronger information (e.g., surrogate OOD data) can also benefit our watermarking strategy, making our proposal very general and attractive. We humbly appreciate your concern, and we believe how to incorporate proper information into our watermarking strategy will be an interesting question that requires further exploration.
>
> > Q2. It is better to explain the use of Eq. (6). This optimization procedure seems very important to the method but lacks enough explanation.
>
> A2. Sorry for our unclear description. To find the proper watermark with minimal risk in Eq. (5), we use the first-order gradient update to iteratively update watermark’s elements. However, directly using the gradient direction in feature updates can lead to unstable optimization [1]. Using the signum of first-order gradients instead can largely mitigate this issue [2], motivating our optimization formula in Eq. (6). We will rephrase the related description in our revision to make it clearer.
>
> > Q3. In the free-energy version, how to set the hyperparameter T?
>
> A3. In our experiments, we directly report the results with $T=1$ following [3]. We also follow your kind suggestion to report the cases with other values of $T$ from the candidate set {1,5,10,50,100,500,1000}. Surprisingly, we find that $T=1$ leads to the best results among the candidates, and we summarize the results on the CIFAR-10 dataset as an example in the following table.
>
> |   T   |     1     |   5   |   10  |   50  |  100  |  500  |  1000 |
> |:-----:|:---------:|:-----:|:-----:|:-----:|:-----:|:-----:|:-----:|
> | FPR95 | **25.94** | 27.84 | 27.76 | 28.93 | 28.48 | 28.02 | 31.21 |
> | AUROC | **95.08** | 94.36 | 94.06 | 93.59 | 93.59 | 93.73 | 93.46 |
> |  AUPR | **98.79** | 98.59 | 98.47 | 98.25 | 98.25 | 98.22 | 98.10 |
>
> > Q4. How to analyze the power of watermarking in math?
>
> A4. The power of adding watermarks can be verified by the properties of a fully-connected ReLU network and its expressive power, summarized by the following theorem.
>
> **Theorem 1**. _Given a fixed fully-connected ReLU network $f_0$ with width $d_m$ , for any Lebesgue-integrable function $g:\mathbb{R}^d\rightarrow\mathbb{R}$ and any $\epsilon>0$, if the width $d_m\le d+4$, then there exists a data-dependent watermark $w$ such that
> $\int |g(x)-f_0(x+\boldsymbol{w})|dx < \epsilon.$_
>
> This theorem states that we can approximate any Lebesgue-integrable function $g$ using the fixed network and a data-dependent watermark, which provides a theoretical foundation to optimize our objective well using the watermarking strategy.
>
> > Q5. The results regarding ImageNet benchmark should be moved into the main content instead of the appendix, which can help better demonstrate the watermarking strategy's performance.
>
> A5. Following your kind suggestion, we will move the ImageNet experiments to the main content in our revision, better demonstrating the power of our watermarking strategy.
>
> > Q6. In the whole paper, the data use ID and OOD to represent the ID or OOD data. While the loss functions use IN and OUT to present parts regarding ID and OOD. Keeping both consistencies is a better option.
>
> A6. Following your suggestion, we will use ID and OOD consistently throughout this paper, making our description clearer.
>
> **Revision Plan**
>
> In our revision, we will further discuss the optimization procedure in Eq. (6), improve the readability and clarity of our paper, move the ImageNet experiments into the main content, and use ID/OOD consistency instead of IN/OUT. Thanks again for your constructive comments.
>
> **General Response**
>
> We have addressed your concerns about our paper. If you have more suggestions, please tell us. We will merge them into our revision as well! Please discuss with us in the openreview system. We will try our best to address your further concerns and merge your comments into our revision.

---

> ### Author Response · Authors · 2022-08-05
> **Looking forward to your responses or further suggestions/comments!**
>
> Dear Reviewer WTn8,
>
> We have addressed your initial concerns regarding our paper. We are happy to discuss them with you in the openreview system if you feel that there still are some concerns/questions. We also welcome new suggestions/comments from you!
>
> Best regards,
>
> Authors of #1621

---

> ### Comment · Reviewer_WTn8 · 2022-08-08
> **Response to Authors**
>
> Thanks for the authors' rebuttal.
> I have read it carefully and it solve with my concerns largely.
> So I would keep my rating as accept (7).

---

> > ### Author Response · Authors · 2022-08-10
> > **Thanks for supporting our paper to be accepted.**
> >
> > Dear Reviewer WTn8,
> >
> > Glad to hear that your concerns are addressed well. Thanks for supporting our paper to be accepted.
> >
> > Best regards,
> >
> > Authors of #1621

---

### Official Review · Reviewer_dM3K · 2022-07-11

**Rating:** 6
**Confidence:** 4
**Soundness:** 3 good
**Presentation:** 3 good
**Contribution:** 2 fair

**Summary:**

The paper proposes a method to improve the OOD detection capability of the existing OOD methods. The paper relies on the "reprogramming" property of DNNs and learns watermarks that, when applied, can better separate the ID and OD data as in existing methods. The paper proposes an algorithm to learn these watermarks more effectively. Experimental results demonstrate that the proposed method can improve the OOD performance of the existing OOD detection methods. The paper also includes several qualitative analysis to study various aspects of the proposed method, including ablation studies, and transferability.

**Questions:**

In general, I find the study of reprogramming in the context of OOD interesting. However, I'm concerned about the practicality of the proposed method as mentioned in the weakness comments.

**Limitations:**

The paper does not explicitly mention the limitation but I think it is discussed in some parts. I would appreciate it if the paper could provide a dedicated section to discuss the limitation of the method.

**Strengths And Weaknesses:**

The paper proposes an interesting approach to improve the performance of the existing OOD methods. In general, I find the paper's contribution novel:
1) A nice motivation for the study of reprogramming in the context of OOD. I can see that the proposed method helps improve the separation of ID and OD data.
2) The proposed algorithm seems to be effective in learning the watermarks.
3) The experiments at least demonstrate the core results of the task, and the benefits of the proposed approach.

I, however, also have some concerns/comments:

1) I can see that in practice, we don't have access to the OD dataset. And there's a large variation in the OOD performance as reported in the ablation study. How exactly do we choose the hyperparameters? Will selecting the wrong hyperparameters can even hurt OOD performance. I'm also not sure about the detail of the ablation experiments (I don't seem to be able to find it in appendix). What OD dataset is used?
2) The previous comment is particularly important when the performance improvements of the proposed method saturate when the dataset has more classes or becomes more complex (e.g., cifar100 or imagenet). This makes the proposed method not favorable to be used in practice.

---

> ### Author Response · Authors · 2022-08-02
> **Response to review questions**
>
> We sincerely thank you for your constructive comments! Please find our responses below.
>
> > Q1. Will selecting the wrong hyperparameters hurt OOD performance? How to choose the hyperparameters?
>
> A1. Your understanding is correct. A wrong setup can truly hurt OOD performance. The experimental results can be found in Figure 5, Tables 3-4 in the main content, and Tables 17 to 28 in the Appendix (cf., Supplementary Material). Furthermore, the details about our hyperparameter selection can be found in Appendix C.6, where we follow [1,2]. It is also summarized below for your convenience.
>
> We adopt validation OOD datasets that are separated from the test OOD datasets including iSUN, Places365, Texture, SVHN, and LSUN. We choose the proper $\sigma_1$ from the candidate parameter set {0.0,0.2,0.4,0.6,0.8,1.0,1.2,1.4,1.6,1.8,2.0} and the proper $\rho$ from {0.0,0.02,0.05,0.07,0.1,0.2,0.5,0.7,1.0,2.0,5.0}. For softmax scoring, $\beta$ is chosen from {0.0,0.5,1.0,1.5,2.0,2.5,3.0,3.5,4.0,4.5,5.0}; and for free energy scoring, $\beta$ is chosen from {0.0,0.02,0.04,0.06,0.08,0.1,0.2,0.4,0.6,0.8,1.0}.
>
> > Q2. The previous comment is particularly important when the performance improvements of the proposed method saturate when the dataset has more classes or becomes more complex (e.g., cifar100 or imageNet). This makes the proposed method not favorable to be used in practice.
>
> A2. Thanks for your comments. First, we want to clarify that we have shown how to select the hyperparameters (see A1). Then, we discuss the effectiveness of our methods in complex situations. The saturation of improvement is a common phenomenon that frequently happens in OOD detection [3,4], since many scoring functions fail in complex setups (e.g., large number of classes). Thus, we next justify that our watermarking strategy works in a large-class setup.
>
> **We have conducted experiments on the ImageNet OOD detection benchmark** in the Appendix C.5 (line 625, page 18 in the original submission) to verify that the watermarking strategy still works when we have many classes. The results are reported in Table 16. The results show that our watermarking strategy still improves the performance of OOD detection a lot. Namely, the proposed watermarking strategy is also effective when the dataset has many classes. For example, after adding the learned watermark, the FPR95 is improved from ~52% to ~44% (see Table 16). We will move Appendix C.5 to the main content in our revision to justify the effectiveness when we meet the many-class situation (i.e., in more practical scenarios).
>
> Lastly, we justify the effectiveness of our watermarking strategy when given different scoring functions.
>
> Since the different scoring functions have their own performance limitation in detecting OOD data, we want to test if the watermarking strategy can further boost the performance of different types of scoring functions. **In Appendix C.1, we show other scoring functions with the watermarking strategy (Tables 9-11).** It can be seen that our strategy can **further boost** the performance of other scoring functions. Besides, we also adopt the MaxLogit in OOD scoring [4] (**another type** of scoring function that is better than softmax in the large-class setup) with our learned watermark. Their comparison on the CIFAR-100 dataset is summarized below.
>
> |      MaxLogit     |  w/ watermark | w/o watermark |
> |:-------------:|:---------------:|:---------------:|
> | FPR95 | **68.15** |  72.37  |
> | AUROC | **83.73** |  79.58  |
> |  AUPR | **96.19** |  94.90  |
>
> As we can see, with the learned watermark, the performance of MaxLogit can be further improved as well.
> The above analysis verifies that the watermarking strategy is effective and general in a large-class setup. It can also further improve the performance of many different scoring functions [4].
>
> **Revision Plan**
>
> In our revision, we will clarify hyperparameter selection in our main content and move the ImageNet-based experiments to the main content. Thanks again for your constructive comments.
>
> **General Response**
>
> We have addressed your concerns about our paper. If you have more suggestions, please tell us. We will merge them into our revision as well! Please discuss with us in the openreview system. We will try our best to address your further concerns and merge your comments into our revision.
>
> **References**
>
> [1] Shiyu Liang, Yixuan Li, and Srikant Rayadurgam. Enhancing the reliability of out-of-distribution image detection in neural networks. NeurIPS (2017).
>
> [2] Xuefeng Du, Zhaoning Wang, Mu Cai, and Yixuan Li. VOS: Learning what you don’t know by virtual outlier synthesis. ICLR (2022).
>
> [3] Fort Stanislav, Jie Ren, and Balaji Lakshminarayanan. Exploring the limits of out-of-distribution detection. NeurIPS (2021).
>
> [4] Dan Hendrycks, Steven Basart, Mantas Mazeika, Andy Zou, Joseph Kwon, Mohammadreza Mostajabi, Jacob Steinhardt. Improving and assessing anomaly detectors for large-scale settings. (2022).

---

> > ### Comment · Reviewer_dM3K · 2022-08-05
> > **One more question**
> >
> > Thank you for your response. It addressed most of my previous concerns. For the validation dataset, is there a specific requirement? More specifically, how is it selected and what is its relationship to the training and test datasets?

---

> > > ### Author Response · Authors · 2022-08-05
> > > **Hyperparameter selection using different validation datasets**
> > >
> > > Many thanks for your valuable comments! We will answer your two questions below.
> > >
> > > Actually, there is no particular requirement for the validation sets, such as requiring that validation sets should follow the same distribution as test situations. In our paper, we assume that the OOD validation sets are separated from the OOD test sets (i.e., iSUN, Places365, Texture, SVHN, and LSUN) just for fair comparison [1, 2]. Further, the test datasets differ from both the training situation (watermark training relies only on ID data) and the test situation (the validation and the test sets use different data).
> > >
> > > Here, we echo our claim that "there is no particular requirement for the validation sets" by the following experiments on the CIFAR-10 dataset regarding the softmax scoring (ID classifiers are trained with the CIFAR-10 dataset). Specifically, we consider using (1) **the validation sets used in [1, 2] (i.e., the individual validation set for iSUN, Places365, Texture, SVHN, and LSUN, respectively)**; and (2) **a new validation set: the tiny-ImageNet**. We report the OOD detection performance on the above validation datasets in terms of FPR95 in the following table. The "candidate" rows represent randomly selected sets of hyperparameter setups. We find that no matter which validation datasets are used in this experiment, the optimal one (the last row (optimal) in the following table) is the same as that used in our paper.
> > >
> > > || $\sigma_1$ | $\rho$ | $\beta$ | Validation Set for iSUN  | Validation Set for  Places365 | Validation Set for  Texture | Validation Set for  SVHN  | Validation Set for  LSUN  | tiny-ImageNet |
> > > |:-------:|:------------:|:--------:|:---------:|:-------:|:-----------:|:---------:|:-------:|:-------:|:-------:|
> > > |candidate| 2.0        | 5.0    | 5.0     | 92.50 |   92.75   | 97.80   | 99.10 | 92.55 | 95.05 |
> > > |candidate| 1.6        | 1.0    | 4.0     | 50.15 |   77.85   | 58.40   | 76.65 | 47.50 | 74.50 |
> > > |candidate| 1.2        | 0.5    | 3.0     | 43.90 |   75.75   | 58.10   | 89.15 | 50.70 | 72.70 |
> > > |candidate| 0.8        | 0.1    | 2.0     | 44.75 |   66.20   | 40.80   | 37.00 | 45.00 | 62.50 |
> > > |candidate| 0.4        | 0.05   | 1.0     | 50.50 |   67.00   | 45.05   | 35.15 | 45.60 | 62.90 |
> > > |optimal| 0.4        | 1.0    | 3.5     | **40.60** | **61.15**     | **40.15**   | **29.85** | **40.40** | **58.30** |
> > >
> > >
> > > As we can see, regarding each of the considered validation sets, we will have the same preference in hyperparameters (i.e., $\sigma_1=0.4, \rho=1.0, \beta=3.5$), aligning with the choice in our paper. It means that the hyperparameters are pretty robust to our particular choice of validation setup. Note that the FPR95 reported in the above table is the FPR95 values on six validation sets: iSUN, Places365, Texture, SVHN, LSUN, and tiny-ImageNet, with different hyperparameters.
> > >
> > > We will also merge the above analysis into our paper. If you have more questions regarding our paper, feel free to tell us. We are very happy to discuss them with you here.

---

> > > ### Author Response · Authors · 2022-08-08
> > > **Further Discussion**
> > >
> > > Dear Reviewer dM3K:
> > >
> > > Thanks for your great efforts in reviewing and good questions here. We really hope that our answer can help to clarify. Since the discussion due is approaching, please let us know if anything we could further clarify.
> > >
> > >
> > >
> > > Best regards,
> > >
> > > Authors of #1621

---

> ### Author Response · Authors · 2022-08-07
> **Looking forward to your reply**
>
> Dear Reviewer dM3K,
>
> We have addressed your initial concerns and replied to your further comments (see https://openreview.net/forum?id=6rhl2k1SUGs&noteId=viaC98ZBSa6). We are happy to discuss them with you in the openreview system if you still have some concerns/questions. We also welcome new suggestions/comments from you!
>
> If all of your concerns are properly addressed and you can confirm this with us, we will be very grateful.
>
> Best regards,
>
> Authors of #1621

---

> ### Author Response · Authors · 2022-08-09
> **Looking forward to your reply**
>
> Dear Reviewer dM3K:
>
> Thanks for your great efforts in reviewing and good questions here. We really hope that our answer can help to clarify. Since the discussion due is approaching, please let us know if anything we could further clarify.
>
> Best regards,
>
> Authors of #1621

---

### Author Response · Authors · 2022-08-07
**Looking forward to the response from Reviewer qcRd**

Dear Reviewer qcRd,

We have completed the experiments regarding large-scale models using ViT (**experiments regarding large-scale models**), please see details in https://openreview.net/forum?id=6rhl2k1SUGs&noteId=Mtd4mcWvqrZ . We have also completed the experiments regarding the near-OOD experiments (**experiments regarding more difficult OOD detection tasks**), please see details in https://openreview.net/forum?id=6rhl2k1SUGs&noteId=vThqHlEQTR_ .

Now, we have addressed all of your initial concerns regarding our paper and provided the required experimental results. We are happy to discuss them with you in the openreview system if you still have some concerns/questions. We also welcome new suggestions/comments from you!

Best regards,

Authors of #1621

---

### Author Response · Authors · 2022-08-09
**Revision Update and Summary of Changes**

Sincerely thanks for the constructive suggestions/comments of all the reviewers. We have correspondingly revised the current submission and marked the revision in blue color in the latest submission.

**For Reviewer dM3K**:

The following points have been added to our revision:

- We describe the hyper-parameter tuning strategy in Section 6 (lines 222-235) and Appendix C.7 (lines 684-686), emphasizing the candidate value sets for the considered hyper-parameters and the random search tuning strategy with validation sets separated from the test situations.

- We add the experiments in Appendix C.2 with Table 11 (MaxLogit) to demonstrate the power of watermarking can benefit from better choices of scoring strategies.

The following point will be added to our revision:

- We will adopt the new hyper-parameter tuning strategy with random search and tiny-ImageNet (an OOD dataset). It will substitute our current tuning strategy to further reflect the generality and effectiveness of watermarking.

**For Reviewer WTn8**:

The following points have been added to our revision:

- We further discuss the optimization procedure in Eq. (5) (lines 162-166) to clarify our purpose.

- We list the performance with free energy scoring on CIFAR benchmarks with different choices of T in Table 37, demonstrating the influence of its value on the performance of free-energy scoring-based watermarking.

- We move the ImageNet experiments and the asscoiated discussions to the main content (Tables 3-4 and lines 257-264), better demonstrating the power of our watermarking strategy.

- We use  ID and OOD consistently throughout this paper (instead of IN and OUT in Eq. 4, 9-10), making our description clearer.

**For Reviewer qcRd**:

The following points have been added to our revision:

- We refine the motivation in reprogramming property for OOD detection (lines 31-34) and the heuristics in why our proposed learning framework works well (lines 40-53 and 126-131).

- We describe the hyper-parameter tuning strategy in Section 6 (lines 222-235) and Appendix C.7 (lines 683-686), emphasizing the candidate value sets for all the hyper-parameters and the tuning strategy of random search.

- We add the near-OOD detection experiments (Table 5) and the associated discussion (lines 265-287) in Section 6, revealing that our watermarking can excel at this challenging setting.

- We add the experiments (Table 24) with different backbone models in Appendix C.6, demonstrating that our proposal is general when facing various model architectures.

- We correct the typos (Eq. 3, Eq. 10, Table 7, lines 314-317) that appear in our previous version.

- We move the discussions about test accuracy and masking to Appendix C.1 and C.8.

- We rename the "Ablation Study" section by "Effect of Hyper-parameters."

- We add experiments (Tables 16-19) about "perm" and "rotate" in Appendix C.3, demonstrating the possibilities in using the shifting augmentations for our watermarking strategy.


The following point will be added to our revision:

- We will adopt the new hyper-parameter tuning strategy with random search and tiny-ImageNet (an OOD dataset). It will substitute our current tuning strategy to reflect further the generality and effectiveness of our watermarking strategy.


Best regards,

Authors

---

### Meta-Review · Area_Chair_fr9P · 2022-08-31

**Recommendation:** Accept
**Confidence:** Certain

**Metareview:**

The reviewers agree that the proposed method is interesting and yields good performance.  A number of concerns were raised during the initial round of reviews concerning the rigorousness and completeness of experiments, but these were addressed during extensive back-and-forth between authors and reviewers.


**Award:**

No

---

### Decision · Program_Chairs · 2022-09-14

Accept